# Omics-based construction of regulatory variants can be applied to help decipher pig liver-related traits
Ziqi Ling [1,2] ✉, Jing Li[1,2], Tao Jiang[1], Zhen Zhang[1], Yaling Zhu[1], Zhimin Zhou[1], Jiawen Yang[1], Xinkai Tong[1], Bin Yang [1] ✉ & Lusheng Huang [1] ✉

Genetic variants can influence complex traits by altering gene expression through changes to regulatory elements. However, the genetic variants that affect the activity of regulatory elements in pigs are largely unknown, and the extent to which these variants influence gene expression and contribute to the understanding of complex phenotypes remains unclear. Here, we annotate 90,991 high-quality regulatory elements using acetylation of histone H3 on lysine 27 (H3K27ac) ChIP-seq of 292 pig livers. Combined with genome resequencing and RNA-seq data, we identify 28,425 H3K27ac quantitative trait loci (acQTLs) and 12,250 expression quantitative trait loci (eQTLs). Through the allelic imbalance analysis, we validate two causative acQTL variants in independent datasets. We observe substantial sharing of genetic controls between gene expression and H3K27ac, particularly within promoters. We infer that 46% of H3K27ac exhibit a concomitant rather than causative relationship with gene expression. By integrating GWAS, eQTLs, acQTLs, and transcription factor binding prediction, we further demonstrate their application, through metabolites dulcitol, phosphatidylcholine (PC) (16:0/16:0) and published phenotypes, in identifying likely causal variants and genes, and discovering sub-threshold GWAS loci. We provide insight into the relationship between regulatory elements and gene expression, and the genetic foundation for dissecting the molecular mechanism of phenotypes.

Pigs, as long-time domesticated animals, have become one of the primary meat sources. In 2020, pork maintained the second-highest average per capita consumption among meat products. To address the increasing demand for pork consumption and improve meat quality, it is crucial to efficiently raise pig populations that exhibit desirable performance matched with the specific requirements, such as high growth rate, low fat, and good adaptation to particular environments. The liver is a key metabolic and heat-producing organ that participates in the processing and utilization of energy sources such as glucose, fatty acids, and amino acids[1]. It plays a vital role in growth, fat metabolism, cold adaptation, and other economic traits in agricultural animals. For example, proteins related to antioxidant enzymes and ribosomal proteins can affect the cold adaptation of pigs at high altitudes[2]. Liver vitamin A metabolism affects feed efficiency in pigs[3]. Liver glucose metabolism correlates strongly with protein and lactose concentrations in the milk of dairy cows[4,5]. Therefore,

understanding the genetic basis of liver-related traits would benefit pig production.

Genome-wide association studies (GWAS) identified thousands of genetic variants responsible for important economic traits[6]. Still, the majority of GWAS loci are located in non-coding regions of the genome[7], hampering the identification of causal variants. In humans, likely causal variants that alter gene expression through changes to regulatory elements were prioritized by the integration of eQTLs and H3K27ac QTLs[8], yet comparable efforts in pigs remain lacking. H3K27ac is one of the most widely studied histone modifications due to its predominant deposition in active promoters and enhancers[9,10], and is highly correlated with gene expression. Recent studies have employed H3K27ac to annotate many regulatory elements in pigs, narrowing down the genome regions containing candidate variants associated with complex traits identified by GWAS[11–13]. However, causal variants hiding in these candidate variants and governing

[1]National Key Laboratory for Swine genetic improvement and production technology, Ministry of Science and Technology of China, Jiangxi Agricultural University, NanChang, Jiangxi Province, P.R. China. [2]These authors contributed equally: Ziqi Ling, Jing Li. ✉e-mail: lingziqi8278@163.com; binyang@live.cn; lushenghuang@hotmail.com

the activity of regulatory elements remain to be pinpointed. Moreover, genetic variants can influence complex traits by modulating gene expression with the assistance of changes to regulatory element activity, but the extent to which genetic effects on gene expression through changes to regulatory element activity remain incompletely characterized genome-wide. Thus, it is necessary to identify genetic variants affecting the activity of regulatory elements genome-wide and elucidate their impact on gene expression, which is valuable for exploring the molecular mechanisms underlying pig complex traits.

In this study, we study H3K27ac activity in up to 292 liver samples from a heterogeneous population managed under the same external environment, thus reducing the variance of environmental factors and amplifying the genetic effect. We identify high-quality H3K27ac peaks and super-enhancers, providing abundant regulatory elements in pig liver. Based on the large sample size, we further characterize inter-individual variation in regulatory element activity, facilitating subsequent acQTLs mapping. Combined with DNA and RNA-seq data, from these individuals, we detect expressed genes, acQTLs, and eQTLs for sharing analyses, colocalization, and GWAS fine-mapping. We validated two putative causal variants contributing to H3K27ac signals in independent datasets through allelic imbalance analyses. Noticeably, both variance decomposition and causal inference analyses support a pleiotropic mode, i.e., in the majority of cases, H3K27ac exhibits a concomitant rather than causative relationship with gene transcription. Furthermore, we demonstrate the utility of H3K27ac, acQTLs, and eQTLs in identifying likely functional gene *AKR1A1*, regulatory element, and causal variant 6_165830307 responsible for liver dulcitol levels and unveiling sub-threshold GWAS variants for liver PC(16:0/16:0) levels. To further interpret GWAS loci for phenotypes that may act through the liver, we intersect our datasets with published variants associated with pig diseases and traits, prioritizing liver-related phenotypes by gene-peak pairs. Overall, we provide a unique resource to disentangle the genetic regulations of H3K27ac states and gene expression, which will facilitate the applicability of GWAS in pig breeding.

## Results

### Data description and annotation of regulatory elements

We obtained H3K27ac profiles from 292 pig livers through chromatin immunoprecipitation and sequencing (ChIP-seq). The *Sscrofa* 11.1 genome was used as the reference for mapping, resulting in an average of 27.4 million uniquely mapped reads per sample (88.7% mapping rate, Supplementary Fig. 1a). After peak calling and filtering procedures, we identified an average of 77,947 H3K27ac peaks per sample, with the fraction of reads in peaks averaging 16% (Supplementary Fig. 1b, Supplementary Data 1). The average peak width across all samples was 691 bp, and the frequency distribution of peaks is highly right-skewed (Fig. 1a). All peaks were further merged into 90,991 consensus peaks that occurred in at least three samples. We defined consensus peaks within the 1 kb of transcription start site (TSS) as promoters and the others as enhancers, yielding 16,544 promoters and 74,447 enhancers. In addition, 41% of the H3K27ac peaks were present in introns, nearly one-third of which were the first introns (Fig. 1b). Twenty-three percent and eighteen percent peaks were distributed in distal intergenic and promoter regions, respectively. Nearly half of the peaks were within 10 ~ 100 kb of the TSS of the nearest gene (Supplementary Fig. 1c). To verify the reality of these peaks, we overlapped H3K27ac peaks in this study with those from a previous study[13]. The results showed that 30,169 H3K27ac peaks from this study covered 98.6% (74,865 out of 75,905) of peaks in research conducted by Kern et al.[13]. The remaining 60,822 peaks contained 56,408 (93%) enhancers, and 41,795 (69%) resided in regulatory regions identified by six epigenetic marks in previous studies[13,14], supporting the reliability of the H3K27ac peaks. We then correlated the occurrence percentage of each consensus peak with its abundance measured by FPM (fragments per million). The peak occurrence percentage was significantly positively associated with its abundance (Spearman's correlation, $\rho = 0.809$, $P$-value $< 2.2 \times 10^{-16}$, Fig. 1c). The top 5000 H3K27ac peaks ranked by FPM were used to overlap with promoters and enhancers (Supplementary

Fig. 1d), showing that H3K27ac activities are generally higher in the promoter (75% overlapped) than in the enhancer. Besides, promoters had a greater likelihood of being shared across individuals than enhancers (Fig. 1d).

Super-enhancers are essential in controlling genes that could determine cell and tissue identity[15]. Herein, we identified an average of 1090 super-enhancers per sample, covering an average of 47.2 kb in width (Supplementary Fig. 1e). These super-enhancers are subsequently merged into 2463 consensus super-enhancers that were found in at least three samples (Supplementary Data 2). The biological coefficients of variation (mean = 0.22) for the peaks within consensus super-enhancers were lower (two-sided *T*-test, *P*-value = $1.12 \times 10^{-102}$) than those of regular peaks (mean = 0.23), indicating that the activity of super-enhancer peaks was more stable across individuals (Supplementary Fig. 1f). Among the 2463 consensus super-enhancers, 43 were active in at least 99% of individuals and covered 237 genes. Analysis of gene enrichment revealed their involvement in liver-related pathways, such as cellular response to lipid (*LDLR, GPBAR1, SCARB1*) and folate metabolism (*MAT1A, SHMT2*), highlighting important roles of these cross-individual shared super-enhancers in maintaining the function of the liver (Fig. 1e, Supplementary Fig. 1g). Taken together, we generated a unique H3K27ac profile of pig liver at the population scale.

To determine the effect of H3K27ac on the transcriptome, we obtained 40 million RNA uniquely mapped reads on average from the same population, with a 98% mapping ratio to the *Sscrofa* 11.1 genome (Supplementary Fig. 1a). A total of 15,509 genes expressed in at least 20% of individuals were identified, 2667 (17%) of which lacked H3K27ac signals in their promoter regions, indicating asynchronous alteration between H3K27ac and gene expression. Using the chromatin accessibility dataset, this analogous phenomenon was also observed in human livers[16]. Enhancer RNAs (eRNAs), transcribed from enhancer regions, play crucial roles in development and disease[17,18]. To determine the putative polyadenylated eRNAs in livers, we selected enhancers from distal intergenic peaks and within 300 bp downstream of genes to assemble new transcripts, harvesting 276 potential eRNAs overlapping with 378 H3K27ac peaks. Notably, 187 (49%) of 378 peaks were located in super-enhancers (Hypergeometric test *P*-value = $1 \times 10^{-22}$), indicating that super-enhancer regions may serve as chromatin niches that facilitate the expression of polyadenylated eRNAs. Compared with ordinary mRNAs, the putative eRNAs had fewer exons and were shorter in length (Fig. 1f and Supplementary Fig. 1h). Fifty-one percent of eRNAs were not spliced (Supplementary Data 3), which is consistent with the characteristics of the eRNAs[19].

### Identification of genetic variants associated with liver H3K27ac

Insights into the inter-individual variation of peak activity and its heritability ($h^2$) can help in understanding pathways from DNA to H3K27ac. We estimated the heritability of peak activity of 88,926 peaks located in autosomes using 278 individuals. The genomes of these individuals were sequenced to an average depth of $7.8 \times$ [20,21]. The activity of regulatory elements is controlled by both cis-QTL and trans-QTL[22]. Thus, we estimated for each peak $h^2_{cis}$ (the variance explained by genetic variants located within ±1 Mb from the peak) and $h^2_{trans}$ (the variance explained by genetic variants located beyond ±5 Mb from the peak). Among 88,926 peaks, 10% peaks have $h^2_{cis}$ greater than 0.2, and 5% peaks have $h^2_{trans}$ greater than 0.2 (Supplementary Fig. 2a, b). Mean $h^2_{cis}$ was 0.064, which was significantly higher than $h^2_{trans}$ (mean = 0.029, two-sided *T*-test, *P*-value $< 2.2 \times 10^{-16}$). We further group $h^2$ into promoters-$h^2$ and enhancers-$h^2$. Estimates of promoters-$h^2$ showed significantly higher than enhancers-$h^2$, regardless of the cis or trans patterns (Supplementary Fig. 2c, d). Besides, estimates of $h^2_{cis}$ are still >$h^2_{trans}$ in both groups (Supplementary Fig. 2e, f).

To further understand the genetic basis underlying the H3K27ac (Supplementary Fig. 3a), we performed H3K27ac quantitative trait loci (acQTLs) analysis based on 30,244,904 single-nucleotide polymorphisms (SNPs) and insertion-deletions (Indels). We represented a cis-acQTL using the lead variant within 1 Mb, and a trans-acQTL as the lead variant >5 Mb from the peak. The results showed that 27% of the 90,991 consensus peaks

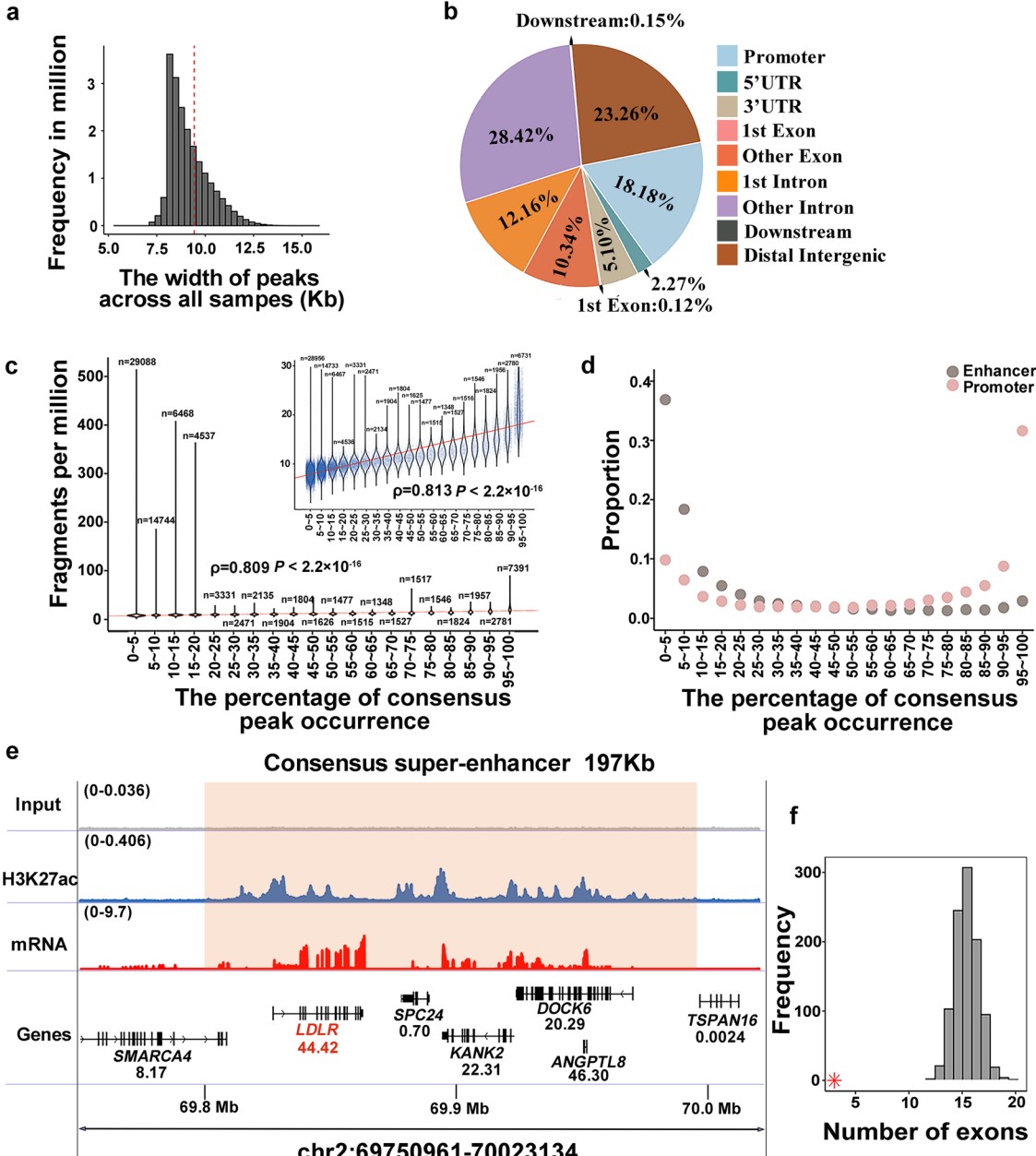

**Fig. 1 | Comprehensive profiling of H3K27ac. a** Distribution of H3K27ac peak width. The red dashed line represents an average width of 691 bp. **b** The category of H3K27ac consensus peak. **c** Correlation between the average fragments per million (FPM, similar to TPM from RNA) of consensus peaks and percentage of its occurrence across individuals. 'n' denotes the number of peaks. Each peak was represented as a blue point. The solid red lines were fitted to the points. The graph in the upper right corner removes outliers by mean ± 3*sd. The correlation and significance were calculated by the Spearman's correlation. **d** The proportion for enhancer or promoter varies with the change in the percentage of consensus peak occurrence. **e** A representative consensus super-enhancer covering lipid-metabolism-related gene *LDLR*. The *x*-axis shows the genomic position. The *y*-axis indicates each base's average read depth (in millions) per 20 bp bin. The input track is the negative control. The transparent orange rectangle highlights the range of the consensus super-enhancer. Gene expression abundances (transcripts per million, TPM) are shown below their symbol. The genomic annotations utilized were sourced from the Ensembl database (version 1.98 of the pig GTF file). **f** The number of exons for polyadenylated enhancer RNAs (eRNAs) and that for genes from reference annotation. The distribution of the number of exons of genes from reference annotation is generated by 1000 permutations. A red asterisk indicates the average number of exons of polyadenylated eRNAs.

were affected by 24,836 cis-acQTLs. Bayesian fine-mapping analyses further revealed that 6651 (27%) cis-acQTLs were narrowed to <200 kb with 95% confidence intervals and 5782 (23%) harbored <20 candidate causal variants (Supplementary Fig. 3b, c). Additionally, we identified 3589 trans-acQTLs responsible for 3395 (3.7%) peaks, 312 of which were associated with trans-chromosome peaks (Supplementary Fig. 3d, Supplementary Data 4 and 5).

To search for the extent of the pleiotropic effect of acQTLs, we conducted pairwise colocalized analysis among lead acQTL variants within a 500 kb distance. The result showed that the majority of acQTLs were

responsible for one peak, which was consistent with findings in humans[23] (Fig. 2a). Eighty-four percent of 27,397 genetic variants included in 28,425 acQTLs were associated with one peak. Notably, we also observed both acQTL 9_118156481 and 14_106813309 were linked to nine peaks on the same chromosome. The nine target peaks for acQTL 14_106813309 are located in the intron of *CYP2C42*, a gene involved in pig liver NADPH-dependent electron transport. These peaks showed high cooperativity of direction (Supplementary Data 4). The majority of acQTLs are cis-acQTLs (87%), which is similar to the result of chromatin accessibility QTLs in

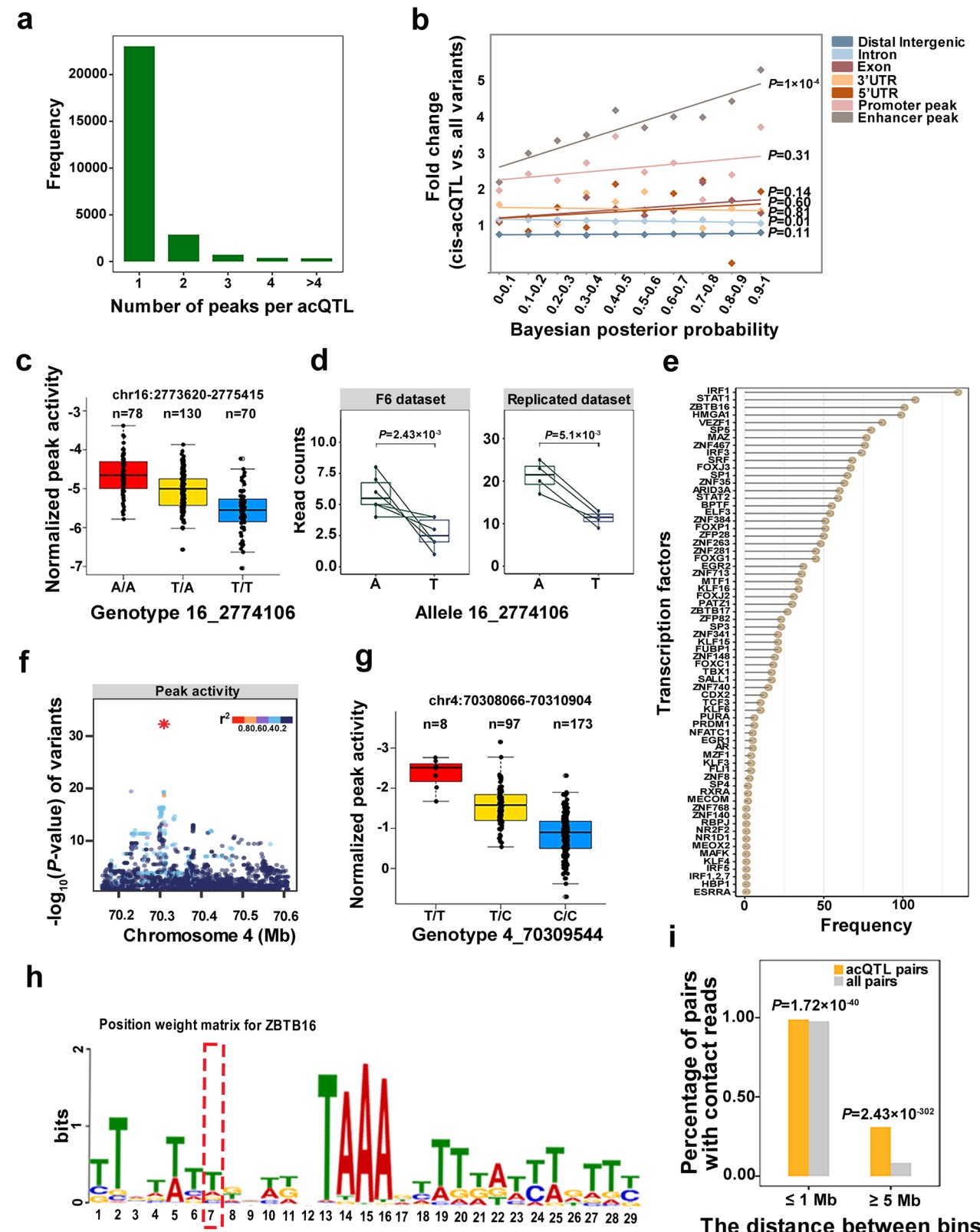

mouse[22]. We then investigated the genomic features of these cis-acQTLs. Among 24,836 cis-acQTLs, 5171 and 19,665 were associated with promoter and enhancer peaks, respectively. We found the cis-acQTLs have a higher enrichment in enhancer peaks than in other genomic features, which is comparable to the result of chromatin accessibility QTLs in human liver[16]. The degree of the enrichment of the cis-acQTLs in enhancer peaks increased with their posterior probabilities (PPs) obtained from the Bayesian fine mapping analysis (Fig. 2b). Cis-acQTLs tend to be symmetrical within 50 kb of their target peaks, and the acQTL closer to the target peaks has a higher association significance (Supplementary Fig. 3d–f), suggesting that genetic variants within and adjacent to the H3K27ac peaks are more likely to regulate the corresponding peaks.

**Fig. 2 | Characterization and transcription factor binding analysis of acQTLs.**
**a** The frequency distribution of acQTLs influencing one or more peaks. **b** The
enrichment of cis-acQTLs varies with alterations in the Bayesian posterior probabilities (PPs). The x-axis represents intervals grouped by the Bayesian PPs. The y-axis represents the enrichment of cis-acQTLs. All variants refer to all genetic variants used for QTL mapping. The solid line fitted the points by a simple linear model. **c** The genotypes of lead acQTL variant 16_2774106 influence the activity of its target H3K27ac. '*n*' denotes sample size. The y-axis represents the normalized activity of the H3K27ac peak that spans the chr16:2773620-2775415 region. **d** The imbalance of the H3K27ac activity at allele 16_2774106 in the F6 and the replicated datasets. The x-axis represents alleles of variant 16_2774106. The y-axis represents read counts covering alleles. Green lines link the same sample. The replicated dataset was downloaded from public databases. The significance of the difference between the two alleles was calculated by two-sided *t*-test. **e** The frequency distribution of binding sites for 67 predicted transcription factors (TFs) whose binding strength was affected

by lead acQTL variants. **f–h** The acQTL chr4:70309544 enhances the binding of TF ZBTB16 when the C allele is changed to T. **f** The association plot for the peak chr4:70308066-70310904. Dots represent variants. The y-axis represents the significance of the association. Lead acQTL 4_70309544 is marked with an asterisk. The colors of variants denote the degree linkage disequilibrium ($r^2$) with the lead acQTL. **g** The genotypes of lead acQTL 4_70309544 influence the activity of peak chr4:70308066-70310904. **h** Position weight matrix (PWM) for the TF ZBTB16. The red dashed box indicates the lead acQTL 4_70309544. The height of bases quantifies the probability of its occurrence. **i** Physical interaction between acQTLs and target peaks. The x-axis represents the distance between bins with a width of 40 kb. The y-axis represents the proportion of bin pairs with Hi-C contacting reads. The significance of control and acQTL pairs was determined by the hypergeometric test. The all pairs refer to all bin pairs with ≤1 M or ≥5 M distance. The boxplots display the median, the 25th and 75th percentiles. The whiskers indicate the minimum and maximum values, and outliers are shown as points outside the ends of the whiskers.

## Exploring the regulatory mechanism of acQTLs

The causal variants determining H3K27ac can contribute to the allelic imbalance of histone marks in heterozygotes[24,25]. To validate the causality of these lead acQTL variants, we examined allelic imbalance of H3K27ac activity for 21 lead acQTL variants that met the following criteria: (1) reads covering the variants have no mapping bias[26]; (2) a sufficient number of heterozygous samples for the statistical test; (3) the variants are located inside H3K27ac peaks; (4) the PPs of variants exceeding 0.9. We observed that 14 lead acQTL variants exhibited consistency between acQTL analysis and allelic imbalance analysis in terms of effect allele direction, eight of which showed significant differences between reads coverage of reference alleles and alternative alleles (Supplementary Data 6). To confirm these 8 lead acQTL variants further, we retrieved 24 H3K27ac data of pig livers from three independent studies[11–13]. Five out of 8 lead acQTL variants had sufficient heterozygous individuals for the statistical test. Four lead acQTL variants showed a consistent tendency, and 2 were successfully verified (Supplementary Data 6, Fig. 2c, d, Supplementary Fig. 3g, h), supporting the reliability of our identified lead acQTL variants.

Genetic variants could regulate the histone modification with the assistance of transcription factors (TFs)[24,27]. We selected 3228 lead acQTL variants inside their target peaks to examine the binding ability of TFs. The results showed that 1288 (39.9%) loci harboring lead acQTL variants could bind TF, and 722 of which were inferred to gain/loss or alter the binding of 67 TFs when alternative alleles substituted reference alleles (Fig. 2e, Supplementary Data 7 and 8). The most frequently affected TFs included IRF1, STAT1, ZBTB16, HMGA1, and VEZF1. For example, the T allele at 70,309,544 on chromosome 4 (4_70309544), associated with greater peak activity of chr4:70308066-70310904, was inferred to have enhanced DNA binding strength with TF ZBTB16 (Fig. 2f–h). The analysis provided a list of candidate TFs regulating the H3K27ac peak activity by binding with lead acQTL variants and partly clarified the regulatory mechanism of H3K27ac.

A pioneering study shows that genetic variants could control distal H3K27ac through spatial interaction in lymphoblastoid cell lines[23]. Herein, we employed Hi-C data from pig liver from another independent research to explore the mechanism of genetic variants affecting H3K27ac of pig liver[28]. Combined with the H3K27ac data of this study, we found that 25,612 (90%) of 28,425 acQTLs contact with their target peaks inferred from the Hi-C data (Supplementary Data 9). Compared with all interaction pairs with matched distance, cis-acQTLs (Hypergeometric test P-value = $1.72 \times 10^{-40}$) and trans-acQTLs (Hypergeometric test P-value = $2.43 \times 10^{-302}$) were significantly enriched in the contact regions encompassing their target peaks (Fig. 2i). Notably, 15% (48 out of 312) inter-chromosome acQTLs contacted genomic regions encompassing their target peaks, supported by an average of 45 Hi-C reads (Supplementary Data 9). Consequently, the Hi-C data not only strengthened the confidence of acQTLs but also agreed with the report where inter-chromosome coordination between regulatory elements had been identified using H3K27ac and Hi-C data in human lymphoblastoid and fibroblast cell lines[29].

## Identification and characterization of eQTLs

H3K27ac is usually linked to the upregulation of gene expression. To explore the relationship between the genetic regulations on gene expressions and that on H3K27ac activity, we identified liver eQTLs in 256 individuals using the same strategies as those used for acQTLs (Supplementary Fig. 4a). The 10,078 (65%) out of 15,509 expressed genes (eGenes) were associated with 12,250 eQTLs, including 10,042 cis-eQTLs and 2208 trans-eQTLs (Supplementary Data 10 and 11). The majority of eQTLs (92%) were found to be associated with one gene (Fig. 3a). Bayesian fine-mapping revealed that 2766 (27%) cis-eQTLs were narrowed to <200 kb with 95% confidence intervals and 2659 (26%) harbored <20 candidate causal variants (Supplementary Fig. 4b). Similar to acQTLs, the majority of eQTLs were located in intron (53%) and distal intergenic (34%; Supplementary Fig. 4c). Moreover, the number of eQTLs and acQTLs across chromosomes is positively correlated (Supplementary Fig. 4d). Comparing the distribution of eQTLs and acQTLs within 2 Mb windows across the entire genome, we discovered a low similarity (Pearson's $R^2 = 0.32$, P-value < $2.2 \times 10^{-16}$) between eQTLs and acQTLs at the genome distribution (Supplementary Fig. 5). We further focused on acQTLs associated with promoter peaks (promoter-acQTLs) and discovered a medium similarity (Pearson's $R^2 = 0.61$, P-value < $2.2 \times 10^{-16}$) between eQTLs and promoter-acQTLs.

Next, we investigated the genomic features of the eQTLs. Among 10,042 cis-eQTLs, 2967 (29.5%) are located within the H3K27ac peaks, preferentially in promoter peaks (Fig. 3b), e.g., 1304 (44.0%) of the 2967 cis-eQTLs were located in the promoter peaks, corresponding to a fold enrichment of 4.02 (Hypergeometric test P-value = $1 \times 10^{-388}$) and the fold change value increased with PPs. In addition, higher PPs were associated with a higher frequency of cis-eQTLs in the promoter peaks of their target genes (Supplementary Fig. 4e and Data 12). Analogous to acQTLs, eQTLs tend to localize within a genomic distance of 50 kb from the TSS (Supplementary Fig. 4f, g), and their association significance was positively associated with the distance from the TSS of target genes (Supplementary Fig. 4h). The results indicated that the genomic proximity between eQTLs and the TSS of target genes is a critical determinant for the likelihood of variants exerting an eQTL effect.

In summary, these analyses revealed similar genomic features of the acQTLs and eQTLs, intriguing the further exploration of the shared genetic controls on H3K27ac activity and gene expression.

## Joint analyses of H3K27ac and transcriptome

Connecting H3K27ac peaks to their target genes is challenging, but promoters are highly associated with gene expression. We examined the sharing between the loci for H3K27ac promoter peaks and those for corresponding gene expression using the $\pi_1$ value (Methods). The majority ($\pi_1 = 0.84$) of promoter-acQTLs were preserved in eQTLs regulating gene expression, which was in line with the result in humans[30]. Approximately half ($\pi_1 = 0.54$) of eQTLs were replicated in promoter-acQTLs, suggesting a substantial sharing of the genetic controls between gene expression and promoter H3K27ac activity (Fig. 3c). Besides, QTL significance for both

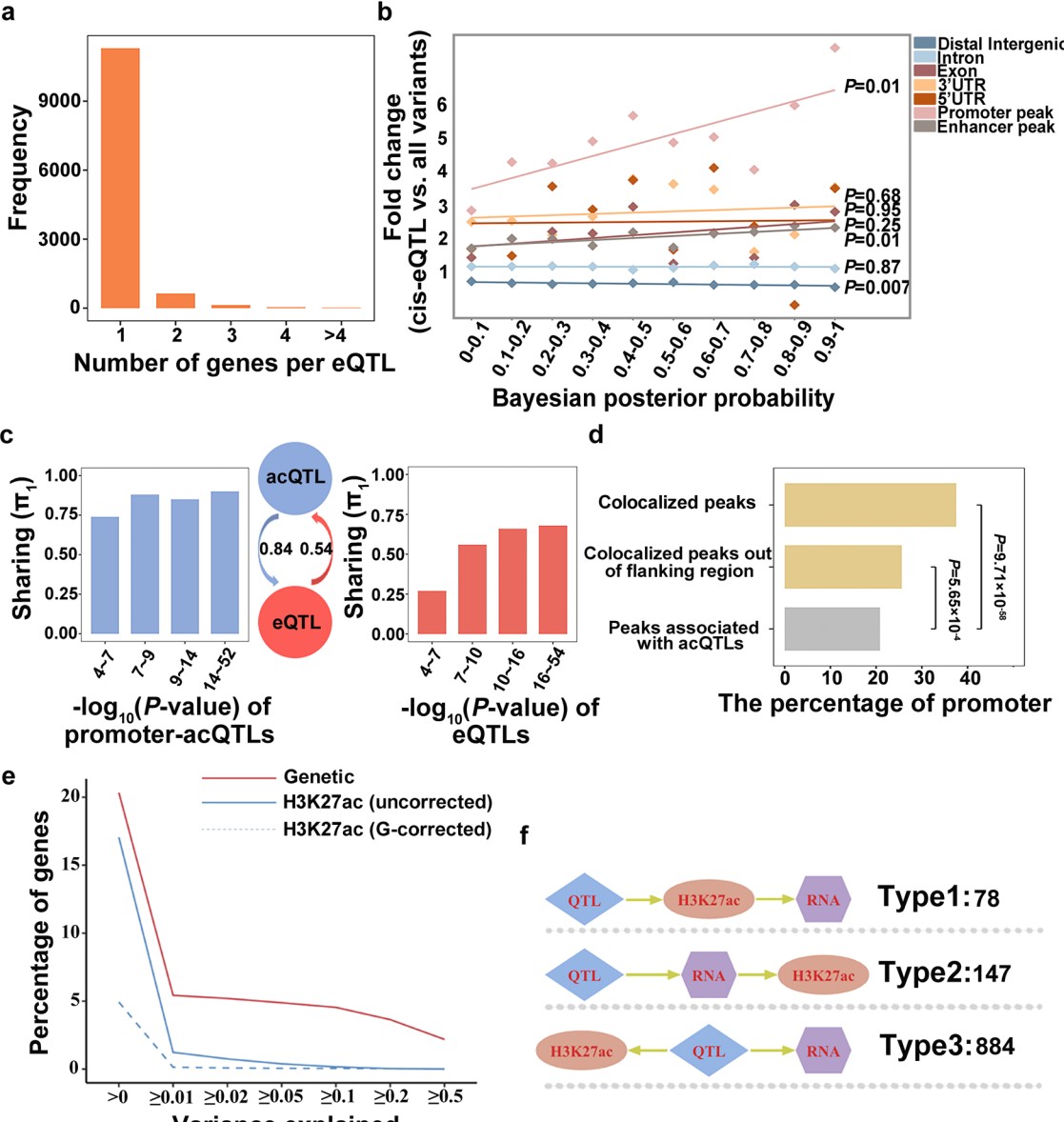

**Fig. 3 | Genetic sharing and causal relationship between H3K27ac and gene expression. a** The frequency distribution of eQTLs influencing one or more genes. **b** The enrichment of cis-eQTLs varies with alterations in the Bayesian posterior probabilities (PPs). The x-axis represents intervals grouped by the Bayesian PPs. The y-axis represents the enrichment of cis-eQTLs. All variants refer to all genetic variants used for QTL mapping. The solid line fitted the points by a simple linear model. **c** QTL sharing between promoter peaks and their corresponding genes. The y-axis displays the sharing score indicated by $\pi_1$. The x-axis represents the intervals of P-values of acQTLs or eQTLs. The intervals were grouped based on the quartile of the P-values. Arrows indicate the direction of the sharing. The sharing from eQTLs to acQTLs is 0.54 for $\pi_1$. The sharing from acQTLs to eQTLs is 0.84 for $\pi_1$. **d** Enrichment of promoter regions within the peaks that have putative target genes. Peaks associated with acQTLs that were colocalized with eQTLs are called colocalized peaks (peaks having putative target genes). Colocalized peaks out of the flanking region refer to colocalized peaks located outside the promoter and gene body of the putative target gene. The x-axis represents the proportion of promoters in all peaks. The P-value was determined by the hypergeometric test. **e** The decomposition of gene expression variance into genetic variants and H3K27ac activity components. The x-axis is the proportion threshold for variance explained by distinct components. First, the decomposition model was independently applied to genetic variants (solid red line) and H3K27ac activity (solid blue line). To modify genetic impacts, the decomposition model then incorporated H3K27ac activity and genetic variants within 100 kb of H3K27ac peaks (blue dashed line). **f** Inference of causation between H3K27ac and their putative target genes. Type1: QTLs impact gene expressions through H3K27ac; Type2: QTLs influence H3K27ac via gene expressions; Type3: QTLs control H3K27ac and gene expression independently. The number of events is denoted at right.

H3K27ac and gene expression led to an increase in the sharing value (Fig. 3c). To determine all shared loci, we then conducted a colocalization analysis. The lead eQTL variants and the lead acQTL variants were considered colocalized when they were in high linkage disequilibrium (LD, $r^2 > 0.8$) and <500 kb apart. The results showed that 2336 acQTLs were colocalized with 1770 eQTLs, leading to 2682 peak-gene pairs (Supplementary Data 13). To reduce the false discovery rate, we kept peak-gene pairs in which the peak is the promoter peak of this gene, or the peak activity

is significantly associated with gene expression by the Spearman's correlation method after multiple testing correction of Benjamini-Hochberg (adjusted P-value < 0.05). Besides, we employed the Bayesian test to perform colocalization and intersected the above results[31]. In total, we identified 1183 target genes for 1616 peaks, comprising 1818 unique peak-gene connections. Most (90%) of the peaks were linked to one gene. But we also identified peaks that were linked to multiple genes. For example, the H3K27ac peak at chr6:54331130-54345331, classified as a promoter of gene *HRC*, was linked

to 4 genes (*HRC*, *CPT1C*, *EMC10*, *FCGRT*). We further checked their physical contact using Hi-C data and found that the *HRC* promoter contacts with gene *CPT1C*, *EMC10*, and *FCGRT*, supported by 23, 4, and 24 Hi-C reads respectively, suggesting promoters can function as enhancers for target genes they interacted with[32] (Supplementary Data 13). The 1616 peaks were significantly enriched at the promoter with a fold enrichment of 1.8 (Hypergeometric test *P*-value = $9.71 \times 10^{-58}$, Fig. 3d). We showed an example of how an acQTL affects the promoter H3K27ac activity and the expression of gene *FLRT3* (Supplementary Fig. 6a–f). Notably, 606 peaks outside the promoter or gene body of target genes appeared to be enriched in the promoters of other genes (Fold enrichment = 1.2, hypergeometric test *P*-value = $5.65 \times 10^{-4}$, Fig. 3d).

To explore the causal hierarchies between H3K27ac and gene expression, we first applied a variance decomposition model to estimate the contribution of genetic and H3K27ac factors to transcriptional variance. The H3K27ac explained a lower proportion of transcriptome variance in models where epigenetic elements were adjusted for proximal genetic effects compared to the corresponding unadjusted models, indicating the correlations between the H3K27ac activity and gene expression for the most part can be attributed to genetic variation (Fig. 3e). Secondly, using QTLs as instructors, we inferred the causal relationships between the H3K27ac and target gene expression with the Intersection-Union Test[33]. It discriminated four causal scenarios: (1) type0, the causal relationship could not be dissolved; (2) type1, QTLs act on gene expression through H3K27ac; (3) type2, QTLs act on H3K27ac through gene expression; (4) type3, QTLs act on H3K27ac and gene expression, respectively. Among 1900 QTL-peak-target gene trios, we identified 791 type0, 78 type1, 147 type2, and 884 type3 scenarios (Fig. 3f), suggesting that a large proportion of H3K27ac accompany rather than determine the gene expression[34,35].

## Identification of functional regulatory elements, genes, and putative causal variants for metabolism-related molecular phenotype and published GWAS loci

Integrating the significant GWAS signals with H3K27ac, gene expression, acQTLs, and eQTLs could aid in identifying functional genes, regulatory elements, and causal variants responsible for interesting traits or diseases[16,36]. Dulcitol is a type of sugar alcohol produced by the reduction of galactose, the excessive accumulation of which could lead to the development of galactosemic cataracts, neurological impairment, and renal dysfunction[37–39]. We first performed GWAS for the liver dulcitol levels using 321 individuals from the same population used for the H3K27ac ChIP-seq, which revealed a notable signal on chromosome 6 with the top SNP 6_165846829 located within an intronic region (Fig. 4a). To identify the regulatory elements regulating dulcitol, we found 19 H3K27ac peaks associated with acQTLs within flanking 500 kb centered at SNP 6_165846829. Moreover, peak chr6:165828531-165836912, which was associated with the acQTL 6_165829987, is the nearest peak to the SNP among the peak-SNP pairs in which the acQTL was colocalized ($r^2 = 0.93$, PP4 = 0.935) with the SNP 6_165846829. We then colocalized acQTL 6_165829987 with eQTLs to link functional genes. The result showed that acQTL 6_165829987 was colocalized with eQTL 6_166100952 ($r^2 = 0.96$, PP4 = 0.99) responsible for gene *AKR1A1* (Aldo-Keto Reductase Family 1 Member A1) and *PRDX1* (Peroxiredoxin 1). Besides, the eQTL 6_166100952 was colocalized ($r^2 = 0.91$, PP4 = 0.96) with the top SNP 6_165846829 from GWAS (Fig. 4b). Notably, *AKR1A1* encodes an Aldo-Keto reductase that is involved in the metabolism of dulcitol[40]. Both the activity of peak chr6:165828531-165836912 and the expression of *AKR1A1* were significantly associated with dulcitol levels (Fig. 4c). Thus, we considered *AKR1A1* and chr6:165828531-165836912 to be a strong candidate gene and regulatory elements, respectively, in controlling dulcitol. To identify likely causal variants, candidate variants were defined within the 95% confidence intervals of the GWAS, the acQTL for peak chr6:165828531-165836912, and the eQTL for gene *AKR1A1* (Fig. 4d). Six variants were collected, among which Indel 6_165830307 not only showed high LD with lead variants for the above three molecular traits but also was located within the peak chr6:165828531-165836912 (Fig. 4d, e).

Furthermore, we predicted the alteration of TF binding when the alternative allele at Indel 6_165830307 substituted the reference allele. Consequently, the binding ability of TFs TP73, TP63, and TP53 were inferred to change significantly (Fig. 4f). We herein highlighted Indel 6_165830307 as a likely causal variant for dulcitol.

Utilizing epigenomic data allows for sub-threshold loci detection in GWAS[41,42]. We first performed GWAS for PC(16:0/16:0) using 321 individuals from the same population used for the H3K27ac ChIP-seq and identified two variants (13_160375595, 13_160375587) exceeding the empirical threshold (*P*-value < $5 \times 10^{-8}$; Supplementary Fig. 7a). To discover more loci, we harvested 4065 nominal variants by employing a weaker threshold with a *P*-value of $1 \times 10^{-4}$. These variants were significantly enriched within the H3K27ac peak region (Hypergeometric test *P*-value = $8.7 \times 10^{-4}$), implying their potential function and stimulating further investigation. We grouped these 4065 nominal variants using LD (minimum $r^2 = 0.2$) to identify 602 independent sub-threshold loci. Among 602 loci, 194 overlapped with H3K27ac peaks, and variants within these loci exhibited significantly stronger *P*-values than the remaining 408 loci (two-sided *T*-test, *P*-value = $3.0 \times 10^{-8}$; Supplementary Fig. 7b). We linked H3K27ac peaks with these 194 loci by colocalizing them with acQTLs through high LD ($r^2 > 0.8$), resulting in 13 loci associated with 13 peaks. Particularly, a locus located at 79.5 Mb on chromosome 13 (locus A) was associated with peak chr13:110306820-110312485 and chr13:110312491-110318835 (Supplementary Fig. 7c–e). Unexpectedly, another independent locus located at 110.2 Mb on chromosome 13 (locus B) was also linked to these two peaks (Supplementary Fig. 7c, e). To identify putative target genes for these two peaks, we searched the previously generated list of peak-gene pairs. The two peaks were discovered to be associated with the gene *PLD1* (Phospholipase D1), which encodes a PC-specific phospholipase involved in PC metabolism[43], and were located within the first intron of this gene (Supplementary Fig. 7c,e). On locus B, the colocalization results between signals of PC(16:0/16:0) and QTLs of three molecular phenotypes were further assessed, i.e. chr13:110306820-110312485 (PP4 = 0.84), chr13:110312491-110318835 (PP4 = 0.84) and *PLD1* (PP4 = 0.84). On locus A, only chr13:110306820-110312485 (PP4 = 0.86) and *PLD1* (PP4 = 0.84) were confirmed. In addition, PC(16:0/16:0) was significantly associated with peak chr13:110306820-110312485 and *PLD1* expression (Supplementary Fig. 7f). Therefore, we proposed *PLD1* as the functional gene and peak chr13:110306820-110312485 as a regulatory element in modulating PC(16:0/16:0).

To further interpret GWAS loci for phenotypes that may act through the liver, we obtained GWAS variants pertaining to 11 categories of pig traits from the ISwine database[44]. We linked 55 phenotypes to 111 gene-peak pairs via LD score ($r^2 > 0.8$), resulting in 167 candidate variants (Fig. 5a, Supplementary Data 14 and 15). The liver is the primary organ responsible for promoting rapid erythrocyte elimination and iron recycling[45,46]. Correspondingly, we found that the variant 7_32054693 for mean corpuscular hemoglobin concentration (MCHC) was linked with the gene *BNIP5* (BCL2 Interacting Protein 5) and its promoter peak chr7:32045442-32062909, which overlapped with the super-enhancer chr7: 32045738-32067174 and was positively associated with the expression of *BNIP5* (Fig. 5b, c). Notably, the gene *BNIP5* was documented in the MGI (Mouse Genome Informatics) dataset as influencing the hematopoietic system. Thus, we proposed that the gene *BNIP5* and the peak chr7:32045442-32062909 may have a functional role in the modulation of MCHC. Besides, variant 7_32054693 was prioritized as a candidate functional variant due to its location within the peak region and inclusion in the 95% credible sets of both the acQTL and the eQTL (Fig. 5d). Similarly, another prioritized candidate functional variant 14_107869191 for hematocrit and red blood cell count resided within the peak chr14:107868236-107869603 and was the lead acQTL variant for this peak as well as the lead eQTL variant for the gene *TLL2* (Supplementary Fig. 8a–c). In addition, we also prioritized genes involved in the growth of pigs, such as average daily gain, for which 5 genes were collected. One of these (*ENSSSCG00000022032*) was documented in the MGI dataset as mouse growth-related genes (Supplementary Data 14).

**Fig. 4 | Identification of candidate causal variants, regulatory elements, and target genes for dulcitol. a** Manhattan plot of GWAS for dulcitol. The *x*-axis represents chromosomes. The y-axis represents the significance of association measured by -log$_{10}$(*P*-value). The empirical genome-wide significance threshold is set to $5 \times 10^{-8}$. **b** Regional plots for GWAS (dulcitol), peak activity (chr6:165828531-165836912), and gene expression (*AKR1A1*). The *x*-axis represents the genome position. The likely causal variant 6_165830307 is denoted as an asterisk. The colors of points denote the degree of linkage disequilibrium ($r^2$) with the lead variant. **c** The pairwise correlation plot of the three molecular phenotypes. The *x*-axis and the *y*-axis represent the normalized molecular phenotypes. The black lines fit these points. The correlation coefficient and *P*-value are calculated by Spearman's correlation. The sample sizes utilized for the correlation analysis were 250, 252, and 203, respectively. **d** Genome browser views of the genomic region harboring candidate causal variants responsible for dulcitol. The *x*-axis displays the genomic position. The *y*-axis indicates each base's average read depth (in millions) per 20 bp bin. The input track is the negative control. The normalized read depths for input, H3K27ac, and mRNA are denoted in brackets. The GWAS track displays candidate variants within 95% confidence interval. The acQTL track shows candidate variants within 95% confidence interval for peak chr6:165828531-165836912. The consensus peak with a width of 8.381 kb is highlighted by a transparent orange rectangle. The eQTL track depicts candidate variants within 95% confidence interval for the *AKR1A1* gene. Gene expression abundances (TPM) are shown below their symbol. The genomic annotations utilized were sourced from the Ensembl database (version 1.98 of the pig GTF file). **e** The effect of the genotype of Indel 6_165830307 on three molecular phenotypes. The *y*-axis represents the normalized phenotypes. '*n*' denotes sample size. The boxplots display the median, the 25th and 75th percentiles. The whiskers indicate the minimum and maximum values, and outliers are shown as points outside the ends of the whiskers. **f** Position weight matrix (PWM) for TFs TP53, TP63, TP73. The red dashed box highlights the matching sequence for the likely causal variant 6_165830307 (C > CTGTTTGAACA). The last four bases (AACA) induce the binding of TFs TP53, TP63, and TP73.

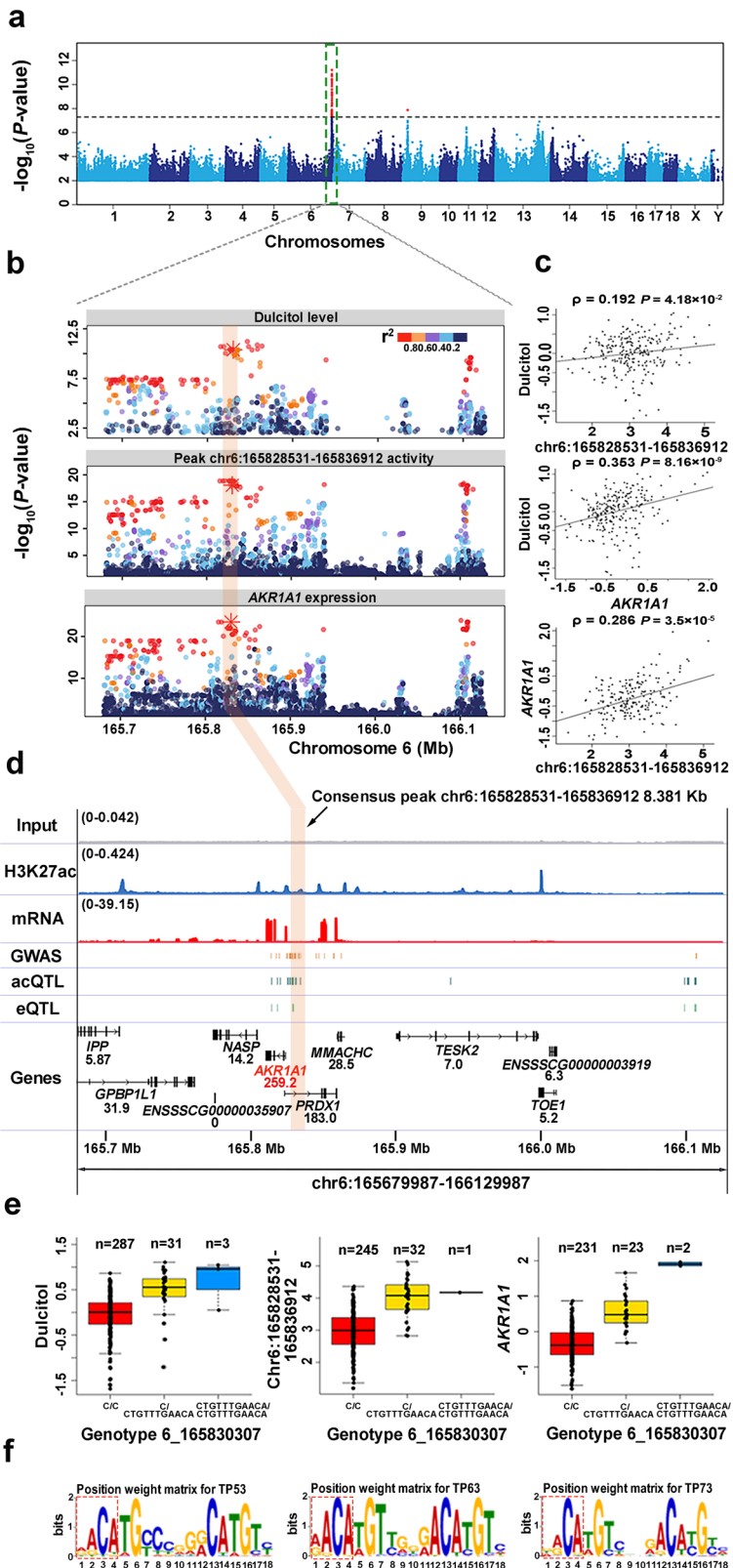

Overall, our analyses thereby demonstrated that acQTLs and eQTLs generated in this study are valuable instruments for dissecting the molecular mechanism of phenotypic GWAS loci.

## Discussion

Epigenetics, as an important regulatory layer, with the assistance of gene expression, has become a powerful tool to dissect molecular mechanisms underlying phenotypic variation. Although many regulatory elements have been annotated by epigenetic marks in pigs[11–13], the effects of genetic variants on epigenetics have yet to be comprehensively characterized. This study identified an extensive set of H3K27ac peaks corresponding to active promoters, enhancers, and super-enhancers. Genetic variants associated with H3K27ac and gene expression were mapped to investigate their relationships and successfully utilized for

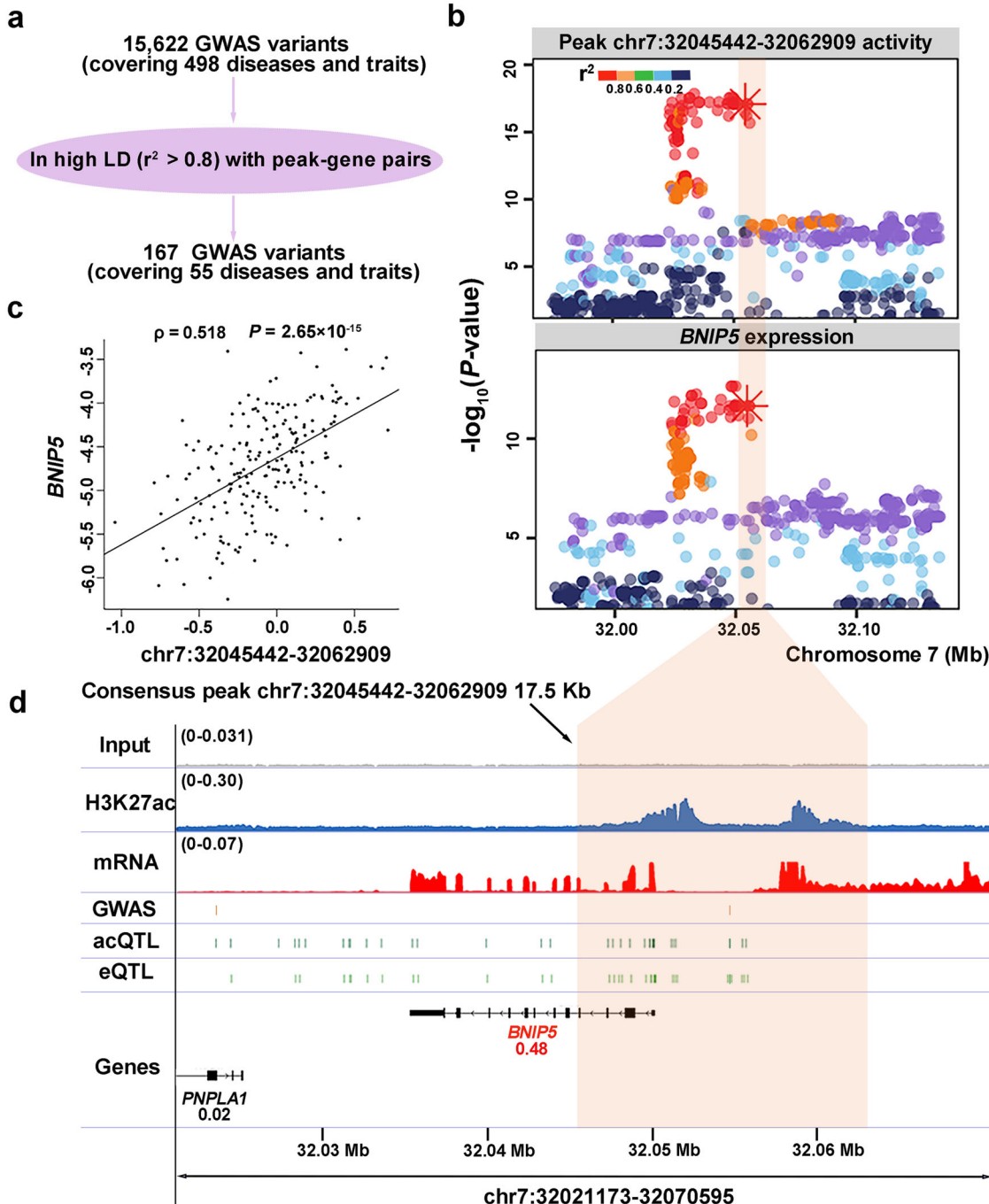

**Fig. 5 | Prioritizing functional variants, genes, and regulatory elements for published GWAS of MCHC. a** Schematic illustrating the workflow for linking GWAS variants to putative functional genes and regulatory elements. **b** Association plot for the peak activity (chr7:32045442-32062909) and gene expression (*BNIP5*). The *x*-axis of the association plot shows the genomic range and the y-axis shows the -log₁₀(*P*-value) for the association analysis. The candidate functional variant 7_32054693 is denoted by an asterisk and the peak chr7:32045442-32062909 is highlighted by a transparent orange rectangle. Dots represent all variants within the genomic range. The pairwise linkage disequilibrium (*r²*) between the lead variant for the corresponding molecular phenotype and all genomic variants is indicated by distinct colors. **c** The correlation plot between the gene expression of *BNIP5* and the peak activity of chr7:32045442-32062909. The *x*-axis and the *y*-axis represent the normalized signals for the peak and the gene, respectively. The solid black line

represents the fitted curve, and the corresponding correlation coefficient and *P*-value are denoted (Spearman's correlation). The sample sizes utilized for the correlation analysis were 203. **d** Genome browser views of the comprehensive molecular profile of the region harboring candidate functional variants for MCHC. The *x*-axis displays the genomic range, and the *y*-axis contains several tracks of data. The GWAS track displays the published variant from MCHC GWAS analysis, and the acQTL track shows candidate variants within 95% confidence interval for the consensus peak chr7: 32045738-32067174 based on fine-mapping analysis. The consensus peak is highlighted by a transparent orange rectangle. The eQTL track depicts candidate variants within 95% confidence interval for the *BNIP5* gene. The TPM values for each gene are displayed in the gene track, with the *BNIP5* gene highlighted in red. The genomic annotations utilized were sourced from the Ensembl database (version 1.98 of the pig GTF file).

aiding in identifying the likely causal genetic variant of liver-related phenotype.

The high overlap of other epigenetic marks from previous studies with H3K27ac peaks in this work shows the credibility of our data. The distribution of liver H3K27ac with respect to genomic features exhibits a similar pattern across pigs, humans, and mouse[8,47]. Previous studies demonstrated that enhancers are less conserved across species and tissues compared to promoters[13,14,48–50]. Our result further showed that enhancers have a lower likelihood of being shared across individuals than promoters. Comparative regulatory genomic analysis in 20 mammalian species has revealed the rapid evolution of enhancers, and the rate of divergence for enhancers is estimated to be 3 times faster than for promoters[48]. Newly evolved enhancers showed high inter-individual variability and tended to be less integrated in transcriptional networks[51]. Indeed, abundant enhancers without contacting promoters do not regulate gene expression[49]. Besides, many enhancers are functionally redundant or have modest effects on target gene expression[52,53]. Consequently, the high variation of enhancers across species, tissues, and individuals might be better tolerated.

Our results indicated a strong peak signal tends to have a high peak occurrence across individuals. There are two reasons for this: (1) strong peak signal is easy to detect, in turn, leading to their high occurrence; (2) regulatory elements with strong peak signals exert an essential function in pig liver, reflected by frequent occurrence. Comparably, promoters and enhancers that are active in both humans and mice have stronger H3K27ac signals than species-specific regulatory elements[54]. We further obtained chromatin states conducted by Pan et al.[14], and found that all-tissue shared promoters/enhancers have higher activity than liver-specific promoters/enhancers (Supplementary Fig. 9). These findings suggested that regulatory elements with high H3K27ac signals tend to be stable across individuals, tissues, and species.

Highly shared super-enhancers cover these genes involved in liver-related pathways, implying their importance in the maintenance of liver function[55]. Combined with RNA-seq data, a significant enrichment of polyadenylated eRNAs within super-enhancer regions corroborates that super-enhancers induce higher levels of eRNAs than typical enhancers[17]. For the 15,509 expressed genes, H3K27ac signals cannot be observed in promoter regions for 17% of genes. This phenomenon has been observed in several studies of accessible chromatin[16,56]. This is possibly a consequence of the limitations of H3K27ac as a mark for completely capturing the accessibility of regulatory elements[57,58]. A previous study also showed that an average of 4.79% accessible pig genome was not marked by any four epigenetic marks including H3K27ac[14]. Another explanation is the potential asynchrony that H3K27ac disappears before mRNA degradation[35].

Using Bayesian fine-mapping analyses, more than 5000 cis-acQTLs with small confidence intervals (<200 kb) and few candidate causal variants (<20) were identified, which could greatly assist in identifying causal variants responsible for interesting traits. The estimates of heritability showed genetic variants located within cis windows are more likely to affect peak activities than trans variants, supporting the result that most acQTLs were cis-acQTLs. Nonetheless, we found 3589 trans-acQTLs, which involved 312 trans-chromosome peaks. Enriching trans-acQTLs and peaks in the Hi-C contact region supports their 3D genomic interactions, consistent with previous studies[23,29]. Besides, acQTLs could be associated with multiple peaks, indicating its pleiotropic effect on H3K27ac. Higher cis-acQTL enrichment in enhancer peaks than promoter could also reflect the rapid evolution of enhancers, possibly due to it harboring more genetic variants. Whether acQTLs are causal mutations or not can be effectively examined by analyzing allelic imbalance of H3K27ac peaks within individuals, due to paternal and maternal alleles functioning as within-sample controls[59]. To explore the regulatory mechanism of acQTLs affecting H3K27ac, the allelic imbalance of H3K27ac activity was successfully validated in several lead acQTL variants, implying the strong reliability of other lead acQTL variants. Furthermore, 1288 lead acQTL variants have binding sites for TF, which not only strengthens the credibility of acQTLs but also implicates its regulatory mechanism.

Our results show that gene expression (or H3K27ac abundance) is more likely to be affected by genetic variants closer to the TSS (or peak midpoint), which also prompts that the causal variants located at the TSS or H3K27ac peak midpoint have a high probability of being causal. High sharing ($\pi_1$) from promoter-acQTLs to eQTLs indicates that gene expression and promoter activity are under the same genetic regulation. The lower $\pi_1$ value in the direction from eQTLs to promoter-acQTLs suggests that RNA regulation may originate from multiple mechanisms, such as RNA processing. The sharing pattern is conserved in different species[30].

In mouse neurons, massively parallel reporter assays demonstrated the sufficiency of promoters to independently initiate transcription while enhancers stimulate transcriptional initiation in a promoter-dependent manner[60]. Analogously, our study also revealed that putative regulatory elements controlling gene expression prefer to be promoters by high enrichment of eQTLs in promoters, similar distribution between eQTLs and promoter-acQTLs, and significant colocalization between promoter-acQTLs and eQTLs.

The variance decomposition showed that genetic factors are the primary determinants of gene expression, in contrast to the view that histone marks play a causal role in transcription. In mouse embryonic stem cells, mutations in genes encoding H3.3 transfer lysine 27 to arginine, preventing H3.3K27 from being acetylated[34]. Despite the dramatically reduced H3K27ac signals in enhancers, enhancer activity remains unchanged and gene expression is barely affected. In another study on K562 cells, rapid loss of H3K27ac was observed after blocking transcription initiation, indicating that H3K27ac serves as a supportive mark in transcription[35]. In the causal inference between H3K27ac and gene expression, 46% of H3K27ac peak-gene pairs that share common QTLs were inferred to be independent, while only 4% supported a causal role of H3K27ac on gene expression. This result supported that a large proportion of H3K27ac is the proxy for regulatory elements rather than the driver.

We identified a likely causal variant 6_165830307 for dulcitol with the help of the resources of this study. The variant 6_165830307 inside peak chr6:165828531-165836912 is significantly associated with dulcitol level and colocalizes with the acQTL for peak chr6:165828531-165836912 and the eQTL for gene *AKR1A1*. When the allele of variant 6_165830307 was altered from the reference allele to the alternative allele, three TFs (TP73, TP63, and TP53) were predicted to bind to the GWAS locus for initial transcription. However, the lack of a sufficient number of heterozygous individuals for variant 6_165830307 hampers further validation through the allelic imbalance of H3K27ac. Taken together, the alternative allele of variant 6_165830307 may increase the activity of regulatory element chr6:165828531-165836912 by enhancing the binding ability of TFs, resulting in the elevated expression of gene *AKR1A1* and high dulcitol level. Our dataset can favor identifying extra GWAS loci with modest effect sizes. In GWAS for PC(16:0/16:0), two independent GWAS loci were linked to the same peaks by acQTLs. *PLD1* harboring the peaks was highlighted as a strong causal gene affecting PC(16:0/16:0). *PLD1* encodes a PC-specific phospholipase involved in PC metabolism[43], indicating its direct relationship with PC(16:0/16:0) phenotype in biological process. Our dataset can also aid in the prioritization of variants and genes for published GWAS signals, such as hematological and growth-related traits, which reinforces the utility of our resource. For example, our result suggests variant 7_32054693 to be a promising candidate for MCHC, likely functioning through the gene *BNIP5* and its promoter peak chr7:32045442-32062909.

Collectively, this study expands the H3K27ac atlas, acQTLs, and eQTLs dataset in pig liver, shedding light on the impact of genetic variants on both H3K27ac and gene expression. This resource will aid in dissecting molecular mechanisms underlying liver-related traits, thus facilitating GWAS fine-mapping.

## Methods
### Ethics statement
All procedures involving animals followed the guidelines for the care and use of experimental animals established by the Ministry of Agriculture of

https://doi.org/10.1038/s42003-024-06050-7                                                                          **Article**

China. The ethics committee of Jiangxi Agricultural University specifically approved this study.

## Samples
All the liver samples were derived from the sixth (F6) generation pigs from a heterogeneous population generated by crossing eight founder breeds including four aboriginal Chinese breeds (Erhualian, Laiwu, Bama Xiang, and Tibetan) and four highly selected international commercial breeds (Duroc, Large White, Landrace, and Pietrain). The population was kept through a rotation mating scheme to acquire an equal mixture of genetic material from eight founder breeds. Feeding conditions were the same for all F6 pigs. Castration was performed on male individuals on day 90. All pigs were slaughtered at 240 ± 10 days of age. Samples were immediately collected and transformed to liquid nitrogen, and then stored at −80 °C until use. Liver samples were obtained from the left lobe of the liver.

## DNA extraction and genotyping
Genomic DNA was obtained from frozen muscle tissue using a phenol-chloroform-based DNA extraction protocol. DNA quality control was performed according to DNA concentrations and length by Nanodrop-1000 and agarose (0.8%) gel electrophoresis. Next, DNA was fragmented into 300-400 bp pieces. After adenylation and indexed ligation, the cDNA library was amplified by PCR using Phusion High-Fidelity DNA polymerase (NEB, USA). The sequencing was completed on Illumina X-10 instruments (Illumina Inc., San Diego, CA) with a 2 × 150 bp paired-end strategy. The low-quality raw reads were removed according to the following criteria: (1) the percentage of N base contents >10%; (2) the percentage of quality score ≤20 bases >50%. After removing low-quality and short reads from raw DNA fastq files, clean fastq files were aligned to the Sscrofa11.1 reference genome using BWA (v0.7.17)[61]. Subsequently, sort and index bam files using Samtools (v1.9)[62]. Individual genotypes were acquired using Platypus (v0.8.1)[63]. VCF files from each sample were merged into a single VCF file using PLINK (v1.9)[64]. Next, imputed the missing genotypes with Beagle (v0.40)[65].

## mRNA extraction and sequencing
Total RNA was isolated using TRIzol™ (Invitrogen, USA) from 256 pig livers, including 146 females and 110 males. The integrity and purity of RNA were tested by an eNanoPhotometer® spectrophotometer (IMPLEN, USA) and a Bioanalyzer 2100 system (Agilent Technologies, USA). Next, mRNA was enriched by poly-T oligo-attached magnetic beads in NEBNext® UltraTMR NA Library Prep Kit for Illumina (NEB, USA). Poly(A) + mRNA was then fragmented and used for strand-specific cDNA library construction. The cDNA was purified, end-repair, A-tailing, adapter ligation, and size selection using AMPure XP beads. The sequencing was performed on the Illumina Novaseq 6000 platform using a 150-bp paired-end strategy. The low-quality raw reads were removed if the percentage of N base contents was >10% or the percentage of Q ≤ 5 bases was >50%.

## ChIP-Seq experiments
Chromatin immunoprecipitation followed by sequencing was performed using the SimpleChIP Plus Enzymatic Chromatin IP Kit (Magnetic Beads) (CST, USA). Pig liver tissue from 172 females and 120 males were collected. In brief, ~200 mg of liver tissue was minced in 1 mL of PBS and cross-linked with 1% formaldehyde for 10 min, followed by quenching with glycine and lysis in the buffer. The cross-linked chromatin was sonicated to produce fragments of 100-300 bp, with 10 μL of the solution reserved as input. The remaining chromatin was then immunoprecipitated with an H3K27ac antibody (active motif, 39133), purified using magnetic beads and a column, and subjected to DNA sequencing with corresponding input samples using an Illumina HiSeq 2500 in a single-end model. The raw reads were filtered to remove reads containing the following: (i) contaminated adapter sequences; (ii) more than half the bases with Phred quality scores below 19; and (iii) >5% ambiguous or undetermined (N) bases.

## RNA-seq data processing
Clean reads were aligned to the Sscrofa11.1 reference genome using STAR (v2.7.1a)[66]. We kept reads with MAPQ value 255 using Samtools Samtools (v1.9). Stringtie (v1.3.6)[67] was used to assemble transcripts with the version 1.98 pig GTF file from Ensembl database using the -e parameter and merged GTF files from each sample into a non-redundant set of transcripts. Quantification of genes was performed using FeatureCounts (v1.5.3)[68].

## ChIP-seq data processing
Clean reads were mapped to the Sscrofa11.1 reference genome using BWA (v0.7.17)[69]. Uniquely mapped reads were obtained using Sambamba (v0.8.1)[70], and duplicates were removed using Picard (v1.119, https://broadinstitute.github.io/picard). To assess library complexity based on ENCODE ChIP-seq Standards, PCR Bottlenecking Coefficient 1 (PBC1), PCR Bottlenecking Coefficient 2 (PBC2), and Non-Redundant Fraction (NRF) were calculated and summarized in Supplementary Data 1. Samples meeting the criteria were used to call peaks with MACS2 (v2.1.1)[71], using input data as the control. The fraction of reads in peaks was calculated for each sample, and a threshold of 1% was utilized to refine the sample set. DiffBind package implemented in R software was used to identify consensus peaks with peaks presenting in at least 3 samples. The reads coverage was calculated using Bedtools (v2.27.0)[72], and peaks were retained if the log2 reads per million (log2RPM) was >0 in at least 3 samples, yielding 91,011 raw peaks. To ensure consistency between acQTL and eQTL mapping analyses, FPM (fragments per million, similar to transcript per million from RNA) was used to represent the activity of H3K27ac peaks, and 90,991 consensus peaks satisfied the further filtering criteria from the GTEx project. The 90,991 peak list was supplied with Supplementary Data 16.

## Quality control for samples
We used verifyBamID (v2.0.1)[73] with Bam files and corresponding genotype files as input to verify sample ID and removed samples when they were predicted to be swapped or contaminated. To eliminate the impact of RNA degradation, RSeQC (v2.6.4)[74] was used to show the coverage profile along the gene body from Bam files, we removed samples that displayed large bias to the 3' end, and left 256 RNA samples manually for subsequent analysis.

## Regulatory elements identified by H3K27ac used for overlapping with that of this study
The predefined regulatory elements using pig liver H3K27ac ChIP-seq data were conducted by Kern et al.[13]. Samples were obtained from two castrated, sexually mature, adult male Yorkshire littermate pigs. The data are available at http://farm.cse.ucdavis.edu/~ckern/Nature_Communications_2020/.

## Chromatin states used for overlapping with regulatory elements of this study
The predefined chromatin states of pigs were obtained from two published independent studies. One was conducted by Kern et al.[13], and they defined 14 distinct chromatin states by utilizing five epigenetic marks (H3K4me3, H3K27ac, H3K4me1, CTCF, H3K27me3) in pig livers. The data are available at http://farm.cse.ucdavis.edu/~ckern/Nature_Communications_2020/. Another was conducted by Pan et al.[14], and they employed five epigenetic marks (H3K4me3, H3K27ac, H3K4me1, ATAC, H3K27me3) to characterize 15 distinct chromatin states in pig livers. The data was available at https://figshare.com/articles/dataset/6_type_of_regulator_hg19_zip/13480425?file=25875270[75].

## Raw sequencing reads for H3K27ac ChIP-seq used for validating the allelic imbalance of H3K27ac
Raw sequencing data for H3K27ac ChIP-seq from 24 pig liver samples were obtained from three independent studies. Two raw data were from Kern et al.[13] and deposited in the Gene Expression Omnibus (GEO) under accession GSE158430. Eight raw data were from Zhao et al.[12] and deposited in the NCBI database under accession number PRJNA597497. Fourteen raw data were from our previous study[11].

## Hi-C matrix

Pig liver Hi-C contact matrix data with 40 kb resolution were generated by Foissac et al.[28], using samples from two male and two female Large White pigs. The data is accessible at the Functional Annotation of Animal Genomes (FAANG) data portal (https://www.fragencode.org/results.html).

## Transcription factors motifs

Position weight matrices of transcription factor binding motifs were collected from the MEME Suite motif database[76], including human, mouse, and mammalian from various sources. The motif matrix data are available at https://meme-suite.org/meme/db/motifs.

## Characterizing H3K27ac peaks

To classify the type of peaks, ChIPseeker (v1.12.1)[77] package implemented in R software was utilized with the version 1.98 pig GTF file from Ensembl database. Specifically, peaks located within the 1 kilobase (kb) range distance from the transcription start site (TSS) were identified as promoters, while those located outside of this range were defined as enhancers. The identification of super-enhancers was performed using ROSE (v1.3.1)[78], and consensus super-enhancers were generated if they present in at least 3 samples using DiffBind (v2.10.0)[79] package implemented in R software. The biological coefficient of variation (BCV) calculated by edgeR (v2.2.6)[80] package implemented in R software was employed to determine the variation of the peaks across the samples. The tagwise dispersion for each peak was calculated.

## Identifying polyadenylated eRNA

Stringtie (v1.3.6)[67] was employed to identify genes without the -e parameter to produce a GTF file without annotated transcripts. Subsequently, genes identified overlapping with distal intergenic and downstream peaks were regarded as candidate enhancer RNAs (eRNAs). To confirm the identity of eRNAs, we performed permutation tests on the annotated genes, based on the number of eRNAs, generating 1000 permutations. Our analysis revealed a P-value of $9.99 \times 10^{-4}$ for both the exon number and length of the eRNAs.

## Heritability estimates for H3K27ac peaks

We employed the GREML-LDMS-I method[81,82] for heritability estimation. For each peak, we estimated cis-heritability using variants within ±1 Mb, while trans-heritability was computed using variants located beyond ±5 Mb. To summarize, we first calculated the segment-based LD score with a 200 Kb window size. SNPs were then stratified into four groups according to LD score quartiles. Following this, GRMs were generated for each SNP group and utilized to estimate heritability.

## Expression QTL (eQTL) mapping

The cis-eQTL analysis was conducted following the Genotype-Tissue Expression (GTEx) project version 8 protocol[83] using the wrapper script. Raw counts matrix and transcripts per million (TPM) values were prepared, and genes with low expression were filtered using the default parameters "--tpm_threshold 0.1 --count_threshold 6 --sample_frac_threshold 0.2". After filtering, 15,509 genes were subjected to the trimmed mean of M-values (TMM) normalization and inverse normal transformation. PEER (probabilistic estimation of expression residuals) factors represent unmeasured and unknown confounders in eQTL mapping, which can be predicted by PEER software (v1.3)[84]. The number of PEER factors was set to 45 as recommended (as detailed at https://github.com/broadinstitute/gtex-pipeline/tree/master/qtl). Covariates such as slaughterAge, transportBatch, RIN_value, gender, uniquely mapped reads, three principal components from genotypes, and 45 PPER factors were adjusted using the limma removeBatch-Effects function (v3.38.3)[85]. The modified FastQTL (v6p)[83] provided by the GTEx project was used with the parameters "--window 1e6 --permute 1000 --maf_threshold 0.01 --ma_sample_threshold 10" to scan the variants within 1 Mb range from TSS and generate a genome-wide empirical P-value threshold for each gene. The empirical P-values were adjusted for multiple testing, and a false discovery rate (FDR)

threshold of 0.05 was used to produce a nominal threshold for each gene. The cis-eQTL was determined by following a fine-mapping analysis for each gene.

Trans-eQTL analysis was performed using QTLtools (v1.3.1)[86] for variants located >5 Mb apart. Permutation was applied with the parameters "--sample 1000" and the false discovery rate (FDR) was set to 0.05 to adjust for multiple testing. The trans-eQTL with the highest level of significance was selected for each gene in each chromosome. To determine the reliability of the trans-eQTL mapping results, we employed a mixed linear model to identify trans-eQTLs with the fastGWA tool[87] in GCTA (v1.9.0) software. The results showed that 99% of trans-eQTLs reach the empirical significance threshold of $5 \times 10^{-8}$, indicating the robustness of trans-eQTLs identification. To eliminate the potential impact of the sex chromosomes, only eQTLs from autosomes were retained.

## H3K27ac quantitative trait loci (acQTLs) mapping

The main analysis was similar to that of eQTLs mapping, and the 90,991 consensus peaks were further utilized as molecular phenotypes. Raw counts matrix and fragment per million (FPM, similar to TPM from RNA) values were prepared, and 90,991 peaks were retained with parameters "--tpm_threshold 0.1 --count_threshold 6 --sample_frac_threshold 0.2". Similar to eQTL mapping, PEER factors were predicted based on H3K27ac signals and the number of factors was also set to 45. Subsequently, covariates such as slaughterAge, transportBatch, gender, uniquely mapped reads, three principal components from genotypes, and 45 PPER factors were adjusted. Cis-acQTL analysis was performed using FastQTL and all variants within a 1 Mb distance from the first base of peaks were utilized. For trans-acQTL analysis, QTLtools were used and all variants outside of the 5 Mb distance were included. Also, only autosomal acQTLs were kept. FastGWA tool further confirmed 99% of trans-acQTLs.

## Fine-mapping analysis

The CAVIAR (v2.2)[88] software was employed for the fine-mapping of variation signals. CAVIAR utilized both the correlation statistical results and the LD information to model and infer the posterior probabilities (PPs) that a variant was causal. The variants were ranked based on their PPs given by CAVIAR in descending order, and the variant sets with cumulative PPs no larger than 0.95 were considered credible variants. To represent a cis-QTL, the variant with the highest PPs was chosen.

## Allelic imbalance analysis of acQTLs

Only acQTLs that were located within the target peaks and had PPs exceeding 0.9 were included. For raw sequencing data for H3K27ac ChIP-seq from other studies, Platypus software was utilized to genotype variants. To mitigate the effects of mapping bias of reads, we employed the WASP (v0.3.4) software developed by van de Geijn et al.[26] to remove reads exhibiting allele-biased mapping. Briefly, WASP was used to flag reads that should be remapped due to overlapping with genetic variants. The alleles of variants were then computationally swapped, and the reads were remapped to determine if they would still be aligned to the original location. After alleles swap, reads with unchanged mapping genome positions were retained. Heterozygous individuals were selected for each acQTL, and the ASEReadCounter[89] function from the Genome Analysis Toolkit (GATK, v.4.2) was employed to quantify allele coverage. Further comparison analysis between the reference and alternative alleles was conducted only on acQTLs that had a minimum of 8 supporting reads and were observed in at least 3 heterozygous individuals.

## Distribution of QTLs

The genomic distribution of all genome variants was analyzed using ChIPseeker package. The variants were classified into seven types based on their location, including 3'UTR, 5'UTR, distal intergenic, exon, intron, promoter peak, and enhancer peak. Cis-QTLs were grouped according to different posterior probability intervals. Cis-acQTLs were further subdivided into two groups, based on their association with either the promoter

or enhancer H3K27ac signals, referred to as promoter-acQTLs and enhancer-acQTLs. The ratio of the seven variant types was calculated for each group, and the fold change values were calculated as the ratio of cis-QTLs divided by the ratio of all variants.

### Physical contact region enrichment analysis

We obtained an interaction matrix from Hi-C with a 40 Kb window size from the pig liver[28] and filtered the interaction bins to include only those with at least 3 supporting contact reads. To avoid confounding effects arising from low contact between different chromosomes, we restricted our analysis to include only acQTLs and target peaks residing on the same chromosome. We calculated the number of contacted bins encompassing acQTLs and target peaks. We then separated all the contact bins into two groups based on distance, i.e., ≤1 Mb and ≥5 Mb, and treated them as control pairs. Finally, we performed a hypergeometric test for both groups.

### Transcription factor prediction

Only acQTLs that were located within target peaks were considered. The primary methodology was derived from a prior study[90]. The process involved extraction of the acQTLs with surrounding 25 bp sequences and utilizing FIMO (v4.11.2)[91] with parameter "—bfile --uniform-- --norc --max-strand" to predict TFBS (transcription factor binding site) on both the reference and alternative sequences. Motif PWMs of TFs from human and mouse were downloaded from the MEME suite website[92]. The impact of acQTLs on TF binding was categorized as either perturbation in binding affinity or loss/gain of binding.

### Estimating sharing of QTLs

To estimate how QTLs are shared between the H3K27ac levels at the promoter and corresponding gene expression, we utilized the $\pi_1$ statistic (qvalue)[93]. In essence, we selected genes with H3K27ac signals in the promoter region that had corresponding acQTLs, calculated the association $P$-value between the promoter-acQTL and gene expression, and estimated the enrichment of low $P$-value via $\pi_1$ estimation. We repeated the same procedure in reverse.

### Variance decomposition of gene expression

We employed a linear mixed model from LIMIX (v2.0.4) to investigate the contributions of genome variation and H3K27ac to gene expression variability[94,95]. The model is as follows:

$$y = N(1\mu, \sigma_l^2 K_l + \sigma_g^2 K_g + \sigma_h^2 K_h + \sigma_e^2 I)$$

Where y represents the gene expression levels across all samples, $1\mu$ represents an offset term, $K_l$ is relatedness matrix built by cis genetic variants or H3K27ac signals, $K_g$ represents a relationship matrix considering all variants and $\sigma_e^2 I$ is the noise term. $K_h$ represents expression heterogeneity and was calculated using the equation $K_h = (1/G)ZZ^T$, in which Z is the $N \times G$ gene expression matrix for N samples and G genes. Subsequently, the proportion of gene expression variability explained by genetic variants or H3K27ac signals was calculated as follows:

$$h = \frac{\sigma_l^2}{\sigma_l^2 + \sigma_g^2 + \sigma_h^2 + \sigma_e^2}$$

In brief, the main variance component considered genetic variants and H3K27ac in our study. Firstly, only the genome variants or H3K27ac within 1 Mb from the gene body were considered independently in the model and the proportion of expression variance explained by them was computed. Next, a joint model across genome variants within 100 kb of H3K27ac peaks and H3K27ac was performed to account for the impact of variants, and the variance explained by H3K27ac alone was calculated.

### Causal inference for H3K27ac and gene expression

We employed the Intersection-Union Test[33] to infer the causal relationship between H3K27ac and gene expression, taking into account genetic information. The causal inference test (CIT) is a mediation-based method introduced by Millstein et al.[33], which examines the hypothesis that a potential causal mediator (G, such as H3K27ac signal) mediates a causal association between a genetic locus (L) and a quantitative trait (T, such as gene expression). Causality (from genetic variants to the mediator to the trait) can be inferred if four conditions are met:

(1) L and G are associated
(2) L and T are associated
(3) L is associated with G, given T
(4) L is independent of T, given G

A total of 1900 candidate L/G/T trios meeting the first two conditions, obtained from peak-gene colocalization analysis, were used for CIT, which can test the strength of a chain of mathematical conditions that as a set are consistent with causal mediation. The Intersection-Union Test framework[33] is used to compute an omnibus $P$-value for the suite of conditions that would function as CIT. For each particular trio with genotype and gene/H3K27ac levels, CIT outputs omnibus $P$-values of a causal model (genetic variants → H3K27ac signals → gene expression; pCausalCIT) and a reactive model (genetic variants → gene expression → H3K27ac signals; pReactiveCIT), which represent the highest $P$-value (i.e., minimal significance) among the four component tests. The CIT predicted casual direction when pCausalCIT <0.05 and pReactiveCIT >0.05 (Type1), and reactive direction when pCausalCIT >0.05 and pReactiveCIT <0.05 (Type2). Trios with pCausalCIT >0.05 and pReactiveCIT >0.05 were considered independent (Type3). The CIT makes no call if pCausalCIT <0.05 and pReactiveCIT <0.05 (Type0).

### Genome-wide association study

The metabolites of pig liver were derived from existing databases of our laboratory. In general, metabolite levels were determined with Ultra-performance liquid chromatography (UPLC) and analyzed with Analyst 1.6.3 software. Covariates for metabolites, such as slaughterAge, transportBatch, and gender, were adjusted using the *lm()* function implemented R program. The simple linear mixed model from Genome-wide Efficient Mixed-Model Association (GEMMA, v.0.97)[96] was employed for further genetic association analyses.

### Colocalization analyses

To search for pleiotropic effects of acQTLs, we calculated the pairwise linkage disequilibrium (LD) score ($r^2$) between lead variants within 500 kb using PLINK (v1.9)[64]. An acQTL is considered to regulate multiple peaks if its LD score with another acQTL is >0.8. The eQTLs were examined identically.

To identify the peak-gene pairs, we calculated the pairwise LD score between acQTLs and eQTLs within a 500 kb range. We obtained peak-gene pairs if the LD score was >0.8. We employed the Bayesian test implemented in COLOC software (v5) to assess colocalization[31]. All variants within a ± 1 Mb from the lead variants of eQTLs and acQTLs were intersected and used for colocalization. The threshold of posterior probabilities of H4 (Association with H3K27ac peaks and gene expression, one shared genetic variant) (PP4) was set to 0.8.

### Integration analysis utilizing published GWAS variants from the ISwine dataset

All GWAS variants for pig phenotypes were obtained from the ISwine dataset[44] (http://iswine.iomics.pro/). The eleven categories of phenotypes were as follows: behavioral, blood, disease, exterior, fat, growth, meat, muscle, physiochemical, reproduction, and slaughter. We removed variants located on sex chromosomes, resulting in 15,622 GWAS variants for 498 phenotypes. In addition, LD scores were calculated between GWAS variants and eQTLs/acQTLs of peak-gene pairs derived from colocalization analysis.

**Article**

Phenotypes were linked to genes or peaks with an LD score threshold of 0.8, resulting in 297 GWAS variants for 64 phenotypes.

## Statistics and reproducibility

Thorough descriptions of the statistical analyses applied in this study are provided in the respective sections of Methods. We utilized 292 samples for peak calling, 256 samples for gene identification, and 321 samples for GWAS. To determine the statistical significance of QTLs associated with peak activity or gene expression, we performed linear regression tests followed by permutation testing using the QTLtools/FastQTLs software suite. H3K27ac ChIP-seq data from 24 pig liver samples were obtained from three independent studies to perform allelic imbalance analysis. To validate acQTLs, t-tests comparing read coverage between alternative and reference alleles were performed, with the resulting P-values used to assess statistical significance. The colocalization among different molecular phenotypes was determined by linkage disequilibrium score, as well as posterior probabilities of H4 from Coloc software. Spearman's correlation was utilized for testing the correlation among different molecular phenotypes.

## Reporting summary

Further information on research design is available in the Nature Portfolio Reporting Summary linked to this article.

## Data availability

All RNA-seq data and ChIP-seq data were publicly available in the GSA database under accession numbers CRA014924, CRA014923 and CRA014930. All genotype data were publicly available at the GVM[20]. Source data for graphs and charts were available at Figshare (https://doi.org/10.6084/m9.figshare.25239307.v1)[97]. GWAS results for the metabolism of phosphatidylcholine (PC) (16:0/16:0) and dulcitol were available at Figshare (https://doi.org/10.6084/m9.figshare.25264963.v1)[98].

## Code availability

The codes for ChIP-seq analysis, RNA-seq analysis, Peak calling, QTL mapping, WASP mapping, Variance decomposition and Casual inference are available from the GitHub repository (https://github.com/lingziqi8278/pig-omics-project/)[99].

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

## Acknowledgements

B.Y. is supported by the National Key Research and Development Program of China (2021YFF1000601). L.H. is supported by the National Natural Science Foundation of China (31790410) and the National Pig Industry Technology System (CARS-35).

## Author contributions

Lusheng Huang: Project administration, Supervision, Funding acquisition, Resources, Writing—review and editing. Bin Yang: Supervision, Data curation, Methodology, Writing—review and editing. Ziqi Ling: Supervision, Formal Analysis, Methodology, Visualization, Writing—original draft. Jing Li: Formal Analysis, Visualization and Methodology. Tao Jiang and Zhen Zhang: Formal Analysis and Methodology. Yaling Zhu and Zhimin Zhou: Methodology. Jiawen Yang and Xinkai Tong: Writing—review and editing.

## Competing interests

The authors declare no competing interests.
