## [Peer review file · Communications Biology]

Reviewers' comments:

Reviewer #1 (Remarks to the Author):

The authors present an extensive resource of H3K27ac, genome sequence and expression data from 292 pig liver samples. They use this data to identify acQTLs and eQTLs and by further integrating them with GWAS data and TF binding prediction show this resource to be a rich resource for the identification of candidate causal variants for phenotypes in the pig. The analyses performed are sound and the results are clearly described. I have mostly suggestions for minor modifications in the text to further improve the clarity of the manuscript, which I have listed below.

Lines 58-61: The first part of this sentence is rather awkward and needs to be rephrased. H3K27ac has not annotated regulatory elements, the authors have used that to do so. This data can then be used to further narrow down regions containing candidate regulatory variants.

Line 61-62: I suggest changing this sentence to: "Thus, it is necessary to identify genetic variants affecting the activity of regulatory elements genome-wide and elucidate their impact on gene expression."

Line 63: Suggest replacing "generate" by "study"

Line 66: Suggest changing this to: "Combined with DNA and RNA-seq data, from these individuals, we"

Line 72: Not clear to what "their" refers to

Line 74: delete "helping to"

Line 79: Delete "s" in "expressions"

Lines 94, 137 and 153: According to the Modern Language Association (MLA), you should never begin a sentence with a numeral. Instead, you should try to reword the sentence. If you can't reword the sentence, spell out the number.

Line 144: Delete "the" before "1 Mb"

Line 151: Change "extend" to "extent"

Line 213: Repalce "cis-acQTLs" by "cis-eQTLs"

Lines 218-219: Suggest changing this to "Comparing the distribution of eQTLs and acQTLs within 2 Mb windows across the entire genome, we discovered a slight similarity". Also consider changing "a slight similarity" as this is a rather vague statement.

Lines 254-255: "indicating the pleiotropic regulatory role of the HRC promoter" The authors should explain/discuss how a promoter of 1 gene can have a pleiotropic effect on 9 other genes.

Line 267: Suggest changing to: "H3K27ac activity and gene expression for most part can be attributed to genetic variation"

Lines 283 and 308: The authors should clearly state that these GWAS analyses were done on the same 292 individuals (at least I assume that was the case).

Line 318: Insert "a" before "locus"

Lines 356-357: Suggest changing this to "Epigenetics, as an important regulatory layer, with the assistance of gene expression, has become a powerful tool to dissect molecular mechanisms underlying phenotypic variation."

Line 358: Replace "abundant" with "many"

Lines 362-369 are just a summary of the methodology used and are better deleted.

Lines 371-372: Delete "Also, we provided thousands of novel regulatory regions for pigs"

Line 373: "There are two reasons for this:"

Line 376: Replace "less" with "lower"

Line 377: "agrees"

Line 382: Replace "In 15,509 expressed genes" by "For the 15,509 expressed genes"

Line 384: Replace "There" with "This"

Line 389: Replace "harvested" with "identified"

Lines 394-395: Delete "But the molecular mechanism is needed to investigate further."

Lines 397-398: I do not understand the statement "The allelic imbalance of histone mark is an effective approach to examine causal owing to other variables being controlled in individual"

Line 399: Replace "were" with "was"

Line 401: Replace "On the other hand" with "Furthermore"

Line 404: Replace "their" with "the"

Lines 427-428: Change sentence to: "However, the lack of a sufficient number of heterozygous individuals for variant 6_165830307 hampers further validation"

Line 438-440: Change sentence to "For example, our result suggests variant 7_32054693 to be a promising candidate for MCHC, likely functioning through the gene BNIP5 and its promoter peak chr7:32045442-32062909."

Line 444: Delete "the"

Line 518: Delete "the" before "RSeQC"

Lines 522-524: These three sentences read rather awkward. I would expect the authors to write what was obtained and where that was obtained and not just give a reference at the end of the sentence.

Line 535: Replace "and produced" with "to produce"

Line 536: Change to "Subsequently, genes identified overlapping with"

Lines 564-565: It is unclear what is meant by "the previously generated reserved peaks"

Lines 576-579. The abbreviation PPs has already been introduced so no need to repeat this. Therefore, also replace "posterior probabilities" with "PPs"

Reviewer #2 (Remarks to the Author):

The manuscript is of great significance for enhancing our understanding of organ-specific

transcription mechanisms. However, I believe that the manuscript requires substantial revisions to improve clarity and flow.

Line 60: The statement here is somewhat unclear. Authors should clarify what the "candidate variants" refer to and their relevance to the study.

The introduction section should clearly state the study's purpose and rationale. Why are the new data and analyses important? What limitations existed in previous studies that this research addresses? Some of the detailed numerical results, which are duplicated in the results section, could be omitted from the introduction.

The manuscript would benefit from incorporating findings from human and mouse studies in the results and discussion sections for comparison. For instance, in Line 158, it would be helpful to know the corresponding percentage in humans or mice. Additionally, in Line 161, it would be valuable to discuss whether the observations are conserved across species.

The discussion should add at least three more paragraphs to discuss the implications and current knowledge in different species. Then readers could understand the importance of new data and analyses.

Overall, these revisions will enhance the manuscript's clarity and make it more informative for readers.

Reviewer #3 (Remarks to the Author):

General comments:

Ling et al. collected large-scale omics data, including WGS, RNA-seq and ChIP-seq of H3K27ac, from 292 pig livers and then explored the genetic control of both H3K27ac activity and gene expression, as well as applied these findings to dissect genetics of complex traits in pigs. The manuscript is well-written, and the data and results are novel and valuable in the field of pig genetics and genomics as claimed by the authors. However, to make the data more impactful and allow others to reproduce the findings, I suggest that all the raw data and the matched metadata should be publicly available. I have a few technical comments and suggestions below, and hopefully they are helpful to improve the manuscript.

Specific comments:

- 1) In figure 1a, it would be better to log-transform the peak width in the x-axis, as it is a highly right skewed distribution. Then you might see that the frequency of peaks is also decreases as their width decreases (Line 89-90).
- 2) Line 250, what is the cutoff of Spearman correlation used to link peaks to genes?
- 3) Before QTL mapping, it would be good to look at the cis/trans-heritability of these H3K27ac peaks and any differences between promoter-H3K27ac and enhancer-H3K27ac.
- 4) For trans-QTL mapping, which covariates have been accounted for, same as cis-QTL mapping? Trans-QTL often have small effects which can be contaminated by artifacts easily (e.g., subtle population structure and cell type composition) like normal GWAS of complex traits. So, it could be better to use much more stringent cutoffs and methods e.g., a linear mixed model, to see if the results are still stable.
- 5) For H3K27ac QTL mapping, how did the authors get 45 PEER factors, same as eQTL mapping, using gene expression rather than H3K27ac signals?
- 6) For allele imbalance analysis of acQTLs, how did the authors take account of mapping bias?
- 7) For variance decomposition of gene expression, did the authors only consider the cis variants and H3K27ac signals, and how the genetic relatedness of individuals was controlled?
- 8) The methods for the causal inference of H3K27ac and gene expression are too simple, e.g., what cutoffs have been used to define significance level.
- 9) For colocalization analysis, there are some often-used methods e.g., COLOC. Did the authors try some of them instead of just looking at LD?

Response to Reviewers

January 15, 2024

We carefully checked the comments and revised the paper by point to point. All revisions were highlighted in red or blue in the manuscript with track changes. The point-by-point responses to the concerns are listed as follows.

Responses to Reviewer #1

The authors present an extensive resource of H3K27ac, genome sequence and expression data from 292 pig liver samples. They use this data to identify acQTLs and eQTLs and by further integrating them with GWAS data and TF binding prediction show this resource to be a rich resource for the identification of candidate causal variants for phenotypes in the pig. The analyses performed are sound and the results are clearly described. I have mostly suggestions for minor modifications in the text to further improve the clarity of the manuscript, which I have listed below.

We thank Reviewer #1 for these very positive comments on the manuscript. We next respond to each of the reviewer's concerns in order.

Lines 58-61: The first part of this sentence is rather awkward and needs to be rephrased. H3K27ac has not annotated regulatory elements, the authors have used that to do so. This data can then be used to further narrow down regions containing candidate regulatory variants.

Responses: We appreciate Reviewer #1 for pointing this out. We have corrected the sentence "Although H3K27ac has annotated abundant regulatory elements in pigs to narrowed down the region containing candidate variants¹⁻³, causal variants governing the activity of regulatory region remain to be pinpointed." to "Recent studies have employed H3K27ac to annotate abundant regulatory elements in pigs, narrowing down the genome regions containing candidate variants associated with complex traits identified by GWAS¹⁻³. However, causal variants hiding in these candidate variants and governing the activity of regulatory elements remain to be pinpointed." (main text: **lines 56-59**).

Line 61-62: I suggest changing this sentence to: "Thus, it is necessary to identify genetic variants affecting the activity of regulatory elements genome-wide and elucidate their impact on gene expression."

Responses: We thank Reviewer #1 for this comment. We have changed the sentence "Thus, it is necessary to identify genome-wide genetic variants affecting the activity of regulatory elements and elucidate their impact on gene expression." to " Thus, it is necessary to identify genetic variants affecting the activity of regulatory elements genome-wide and elucidate their impact on gene expression" (main text: **lines 63-64**).

Line 63: Suggest replacing "generate" by "study"

Responses: Has been corrected (main text: **line 66**).

Line 66: Suggest changing this to: "Combined with DNA and RNA-seq data, from these individuals, we"

Responses: We thank Reviewer #1 for this helpful suggestion. We have changed "Combined with their DNA sequencing and RNA-seq data, we" to "Combined with DNA and RNA-seq data, from these individuals, we" (main text: **lines 71-72**).

Line 72: Not clear to what "their" refers to.

Responses: We thank Reviewer #1 for pointing out this unclear statement. The "their" at Line 72 in the original manuscript refers to H3K27ac, acQTLs, and eQTLs. In the revised manuscript, we have corrected the sentence "Furthermore, we demonstrate their utility in identifying likely functional gene *AKR1A*, regulatory element, and causal variant 6_165830307 responsible for liver dulcitol levels and helping to unveil sub-threshold GWAS variants for liver phosphatidylcholine (PC) (16:0/16:0) levels." to "Furthermore, we demonstrate the utility of H3K27ac, acQTLs, and eQTLs in identifying likely functional gene *AKR1A*, regulatory element, and causal variant 6_165830307 responsible for liver dulcitol levels and unveiling sub-threshold GWAS variants for liver phosphatidylcholine (PC) (16:0/16:0) levels." (main text: **lines 77-80**).

Line 74: delete "helping to".

Responses: Has been deleted (main text: **line 79**).

Line 79: Delete "s" in "expressions"

Responses: Has been deleted (main text: **line 84**).

Lines 94, 137 and 153: According to the Modern Language Association (MLA), you should never begin a sentence with a numeral. Instead, you should try to reword the sentence. If you can't reword the sentence, spell out the number.

Responses: We thank Reviewer #1 for pointing these out. We have reworked the sentence "23% and 18% peaks were distributed in distal intergenic and promoter regions, respectively." to "Twenty-three percent and eighteen percent peaks were distributed in distal intergenic and promoter regions, respectively." (main text: **lines 99-100**), the sentence "51% of eRNAs were not spliced (Supplementary Table S3), which is consistent with the characteristics of the eRNAs." to "Fifty-one percent of eRNAs were not spliced (Supplementary Table S3), which is consistent with the characteristics of the eRNAs." (main text: **lines 142-143**), the sentence "28,425 acQTLs included 27,397 variants, of which 23,025 (84%) were associated with one peak." to "Eighty-four percent of 27,397 genetic variants included in 28,425 acQTLs were associated with one peak." (main text: **lines 172-173**).

Line 144: Delete "the" before "1 Mb"

Responses: Has been deleted (main text: **line 162**).

Line 151: Change "extend" to "extent"

Responses: Has been changed (main text: **line 169**).

Line 213: Replace "cis-acQTLs" by "cis-eQTLs"

Responses: Has been corrected (main text: **line 234**).

Lines 218-219: Suggest changing this to "Comparing the distribution of eQTLs and acQTLs within 2 Mb windows across the entire genome, we discovered a slight similarity". Also consider changing "a slight similarity" as this is a rather vague statement.

Responses: We thank Reviewer #1 for this suggestion. To make this statement clearer, we measured the similarity of their distribution in the genome by Pearson Correlation Coefficient with sliding windows of 2 Mb across the entire genome. The distribution of eQTLs and acQTLs in the genome witnessed a low similarity (Pearson's $R^2=0.32$, P -value $<2.2 \times 10^{-16}$). A medium similarity (Pearson's $R^2=0.61$, P -value $<2.2 \times 10^{-16}$) was observed between eQTLs and promoter-acQTLs at the genome distribution. Thus, we have changed the sentence "Comparing the distribution of eQTLs and acQTLs across the entire genome, we discovered a slight similarity with 2 Mb windows. We further focused on acQTLs associated with promoter peaks (promoter-acQTLs) and discovered a higher degree of similarity." to "Comparing the distribution of eQTLs and acQTLs within 2 Mb windows across the entire genome, we discovered a low similarity (Pearson's $R^2=0.32$, P -value $<2.2 \times 10^{-16}$) between eQTLs and acQTLs at the genome distribution (Supplementary Figure S5). We further focused on acQTLs associated with promoter peaks (promoter-acQTLs) and discovered a medium similarity (Pearson's $R^2=0.61$, P -value $<2.2 \times 10^{-16}$) between eQTLs and promoter-acQTLs." (main text: **lines 239-244**).

Lines 254-255: "indicating the pleiotropic regulatory role of the HRC promoter" The authors should explain/discuss how a promoter of 1 gene can have a pleiotropic effect on 9 other genes.

Responses: We appreciate Reviewer #1 for this instructive suggestion. After considering the comment of Reviewer #3 (please see pages: **23-24** of this response letter), we employed the Bayesian test COLOC⁴ to enhance the robustness of the colocalization analysis. Subsequently, the H3K27ac peak at chr6:54331130-54345331, classified as a promoter of gene HRC, was linked to 4 genes (*HRC*, *CPTIC*, *EMC10*, *FCGRT*). These four genes are close to each other within approximately 1 Mb area (chr6:54337374-55322250) on chromosome 6. We then checked their physical interaction using Hi-C data and found that the *HRC* promoter interacts with gene

CPTIC, *EMC10*, and *FCGRT*, supported by 23, 4, and 24 Hi-C reads respectively. Thus, we speculate that the *HRC* promoter can regulate 3 other genes by physical interaction. Besides, Chandra et al.⁵ demonstrated that promoters were capable of functioning as enhancers for target genes. We have corrected the sentence "For example, the H3K27ac peak at chr6:5433113054345331, classified as a promoter of gene *HRC*, was linked to 4 genes (*HRC*, *CPTIC*, *EMC10*, *FCGRT*). We further checked their physical contact using Hi-C data and found that the *HRC* promoter contacts with gene *CPTIC*, *EMC10*, and *FCGRT*, supported by 23, 4, and 24 Hi-C reads respectively, suggesting promoters can function as enhancers for target genes they interacted with" (main text: **lines 279-283**).

Line 267: Suggest changing to: "H3K27ac activity and gene expression for most part can be attributed to genetic variation"

Responses: We thank Reviewer #1 for this suggestion. We have changed the sentence "H3K27ac activity and gene expression can be majorly attributed to genetic variation" to "H3K27ac activity and gene expression for the most part can be attributed to genetic variation" (main text: **line 295**).

Lines 283 and 308: The authors should clearly state that these GWAS analyses were done on the same 292 individuals (at least I assume that was the case).

Responses: We thank Reviewer #1 for this suggestion. These GWAS analyses were done on the same population used for the H3K27ac ChIP-seq of 292 pigs, but it used 320 individuals. Now, we have corrected the sentence "We first performed GWAS for the liver dulcitol levels, which revealed a notable signal on chromosome 6 with the top SNP 6_165846829 located within an intronic region." to "We first performed GWAS for the liver dulcitol levels using 320 individuals from the same population used for the H3K27ac ChIP-seq, which revealed a notable signal on chromosome 6 with the top SNP 6_165846829 located within an intronic region." (main text: **lines 310-313**). The sentence "We first performed GWAS for PC(16:0/16:0) and identified two variants (13_160375595, 13_160375587) exceeding the empirical threshold (P -value $< 5 \times 10^{-8}$; Supplementary Figure S6A)." was corrected to "We first performed GWAS for PC(16:0/16:0) using 320 individuals from the same population used for the H3K27ac ChIP-seq and identified two variants (13_160375595, 13_160375587) exceeding the empirical threshold (P -value $< 5 \times 10^{-8}$; Supplementary Figure S7A)" (main text: **lines 336-339**).

Line 318: Insert "a" before "locus"

Responses: Has been inserted (main text: **line 348**).

Lines 356-357: Suggest changing this to "Epigenetics, as an important regulatory layer, with the assistance of gene expression, has become a powerful tool to dissect molecular

mechanisms underlying phenotypic variation."

Responses: We thank Reviewer #1 for this suggestion. We have changed the sentence "Epigenetics, as an important regulatory layer, have become a powerful tool to dissect molecular mechanisms from DNA to phenotype with the assistance of gene expression." to "Epigenetics, as an important regulatory layer, with the assistance of gene expression, has become a powerful tool to dissect molecular mechanisms underlying phenotypic variation." (main text: **lines 390-391**).

Line 358: Replace "abundant" with "many"

Responses: Has been replaced (main text: **line 392**).

Lines 362-369 are just a summary of the methodology used and are better deleted.

Responses: We thank Reviewer #1 for this good suggestion. The corresponding sentences have been deleted in the first paragraph of the Discussion section in the revised manuscript. Additionally, we have reworked the first paragraph of the Discussion section as the following: "Epigenetics, as an important regulatory layer, with the assistance of gene expression, has become a powerful tool to dissect molecular mechanisms underlying phenotypic variation. Although many regulatory elements have been annotated by epigenetic marks in pigs¹⁻³, the effect of genetic variants on epigenetics remains unexplored. This study identified an extensive set of H3K27ac peaks corresponding to active promoters, enhancers, and super-enhancers. Genetic variants associated with H3K27ac and gene expression were mapped to investigate their relationships and successfully utilized for aiding in identifying the likely causal genetic variant of liver-related phenotype."(main text: **lines 390-397**).

Lines 371-372: Delete "Also, we provided thousands of novel regulatory regions for pigs"

Responses: Has been deleted. (main text: **line 399**).

Line 373: "There are two reasons for this:"

Responses: Has been corrected (main text: **line 412**).

Line 376: Replace "less" with "lower"

Responses: Has been corrected (main text: **line 402**).

Line 377: "agrees"

Responses: We thank Reviewer #1 for this comment. After considering the comment of Reviewer #2 (please see pages: **14-15** of this response letter), we changed "agree with

the view of newly generated enhancers exhibiting significant inter-individual variation" to "Newly evolved enhancers showed high inter-individual variability and tended to be less integrated in transcriptional networks⁶."(main text: **lines 405-407**).

Line 382: Replace "In 15,509 expressed genes" by "For the 15,509 expressed genes"

Responses: We appreciate Reviewer #1 for this suggestion. It has been replaced (main text: **lines 424-425**).

Line 384: Replace "There" with "This"

Responses: Has been replaced (main text: **line 426**).

Line 389: Replace "harvested" with "identified"

Responses: Has been replaced (main text: **line 433**).

Lines 394-395: Delete "But the molecular mechanism is needed to investigate further."

Responses: Has been deleted. (main text: **line 440**).

Lines 397-398: I do not understand the statement "The allelic imbalance of histone mark is an effective approach to examine causal owing to other variables being controlled in individual"

Responses: We thank Reviewer #1 for this comment. Histone marks (e.g., H3K27ac) may exhibit different abundance at paternal and maternal alleles, which is caused by alterations of the transcription factor motif⁷. Paternal and maternal alleles function as within-sample controls, allowing for the identification of regulatory variants without other confounding factors such as environment or genetic background. Thus, we want to express whether cis-acQTLs are causal mutations or not can be examined by analyzing allelic imbalance of histone marks within individuals. In the revised manuscript, we have corrected this sentence to "Whether acQTLs are causal mutations or not can be effectively examined by analyzing allelic imbalance of H3K27ac peaks within individuals, due to paternal and maternal alleles functioning as within-sample controls⁷." (main text: **lines 442-444**).

Line 399: Replace "were" with "was"

Responses: Has been replaced (main text: **line 446**).

Line 401: Replace "On the other hand" with "Furthermore"

Responses: Has been replaced (main text: **line 447**).

Line 404: Replace "their" with "the"

Responses: Has been replaced (main text: **line 450**).

Lines 427-428: Change sentence to: "However, the lack of a sufficient number of heterozygous individuals for variant 6_165830307 hampers further validation"

Responses: We appreciate Reviewer #1 for this comment. We have changed the sentence "However, the insufficient heterozygous individual for variant 6_165830307 hampers further validation through allelic imbalance of H3K27ac." to "However, the lack of a sufficient number of heterozygous individuals for variant 6_165830307 hampers further validation through the allelic imbalance of H3K27ac." (main text: **lines 478-480**).

Line 438-440: Change sentence to "For example, our result suggests variant 7_32054693 to be a promising candidate for MCHC, likely functioning through the gene BNIP5 and its promoter peak chr7:32045442-32062909."

Responses: We appreciate Reviewer #1 for this suggestion. We have changed the sentence "For example, our result proposed the variant 7_32054693 was a promising candidate for MCHC, and function through the gene *BNIP5* and its promoter peak chr7:32045442-32062909." to "For example, our result suggests variant 7_32054693 to be a promising candidate for MCHC, likely functioning through the gene *BNIP5* and its promoter peak chr7:32045442-32062909." (main text: **lines 490-491**).

Line 444: Delete "the"

Responses: Has been deleted. (main text: **line 495**).

Line 518: Delete "the" before "RSeQC"

Responses: Has been deleted. (main text: **line 571**).

Lines 522-524: These three sentences read rather awkward. I would expect the authors to write what was obtained and where that was obtained and not just give a reference at the end of the sentence.

Responses: We appreciate Reviewer #1 for pointing out this. We have added the corresponding information to the Materials and Methods section in the revised manuscript as follows (main text: **lines 574-606**):

"Public datasets

Regulatory elements identified by H3K27ac used for overlapping with that of this

study

The predefined regulatory elements using pig liver H3K27ac ChIP-seq data were conducted by Kern et al.³. Samples were obtained from two castrated, sexually mature, adult male Yorkshire littermate pigs. The data are available at http://farm.cse.ucdavis.edu/~ckern/Nature_Communications_2020/.

Chromatin states used for overlapping with regulatory elements of this study

The predefined chromatin states of pigs were obtained from two published independent studies. One was conducted by Kern et al.³, and they defined 14 distinct chromatin states by utilizing five epigenetic marks (H3K4me3, H3K27ac, H3K4me1, CTCF, H3K27me3) in pig livers. The data are available at http://farm.cse.ucdavis.edu/~ckern/Nature_Communications_2020/. Another was conducted by Pan et al.⁸, and they employed five epigenetic marks (H3K4me3, H3K27ac, H3K4me1, ATAC, H3K27me3) to characterize 15 distinct chromatin states in pig livers. The data was available at https://figshare.com/articles/dataset/6_type_of_regulator_hg19_zip/13480425?file=25875270.

Raw sequencing reads for H3K27ac ChIP-seq used for validating the allelic imbalance of H3K27ac

Raw sequencing data for H3K27ac ChIP-seq from 24 pig liver samples were obtained from three independent studies. Two raw data were from Kern et al.³ and deposited in the Gene Expression Omnibus (GEO) under accession GSE158430 (<https://www.ncbi.nlm.nih.gov/geo/query/acc.cgi?acc=GSE158430>). Eight raw data were from Zhao et al.² and deposited in the NCBI database under accession number PRJNA597497 (<https://www.ncbi.nlm.nih.gov/bioproject/PRJNA597497/>). Fourteen raw data were from our previous study¹.

Hi-C matrix

Pig liver Hi-C contact matrix data with 40 kb resolution were generated by Foissac et al.⁹, using samples from two male and two female Large White pigs. The data is accessible at the Functional Annotation of Animal Genomes (FAANG) data portal (<https://www.fragencode.org/results.html>).

Transcription factors motifs

Position weight matrices of transcription factor binding motifs were collected from the MEME Suite motif database¹⁰, including human, mouse, and mammalian from various sources. The motif matrix data are available at <https://meme-suite.org/meme/db/motifs>.

Line 535: Replace "and produced" with "to produce"

Responses: Has been replaced (main text: **lines 618-619**).

Line 536: Change to "Subsequently, genes identified overlapping with"

Responses: We appreciate Reviewer #1 for this suggestion. We have changed the sentence "Subsequently, the genes identified that overlapped with distal intergenic and

downstream peaks were regarded as candidate enhancer RNAs (eRNAs)." to "Subsequently, genes identified overlapping with distal intergenic and downstream peaks were regarded as candidate enhancer RNAs (eRNAs)." (main text: **lines 619-620**).

Lines 564-565: It is unclear what is meant by "the previously generated reserved peaks"

Responses: We appreciate Reviewer #1 for pointing out this vague description. "the previously generated reserved peaks" refers to the 90,991 consensus peaks satisfying the filtering criteria of the GTEx project in the section ChIP-seq Data Processing in Materials and Methods. Now, we have corrected the sentence "The main analysis was similar to that of eQTLs mapping, but the previously generated reserved peaks were further utilized." to "The main analysis was similar to that of eQTLs mapping, and the 90,991 consensus peaks were further utilized as molecular phenotypes." (main text: **lines 660-661**).

Lines 576-579. The abbreviation PPs has already been introduced so no need to repeat this. Therefore, also replace "posterior probabilities" with "PPs"

Responses: We thank Reviewer #1 for pointing this out. It has been replaced (main text: **lines 674-675**).

Finally, we sincerely thank Reviewer #1 for reviewing our manuscript and providing insightful comments, which have helped us improve the quality of our work. We hope that the responses provided here addressed all the comments in a satisfactory manner.

Responses to Reviewer #2

The manuscript is of great significance for enhancing our understanding of organ-specific transcription mechanisms. However, I believe that the manuscript requires substantial revisions to improve clarity and flow.

We thank Reviewer #2 very much for reviewing our manuscript and positive assessment. We next address each of the reviewer's comments in order.

Line 60: The statement here is somewhat unclear. Authors should clarify what the "candidate variants" refer to and their relevance to the study.

Responses: We thank Reviewer #2 for pointing out this and apologize for the unclear statement. The "candidate variants" refer to genetic variants associated with pig complex traits identified by GWAS. One of the primary purposes of the study is to prioritize likely causal variants among these candidate variants through H3K27ac. In the revised manuscript, we have changed the sentence "Although H3K27ac has annotated abundant regulatory elements in pigs to narrowed down the region containing candidate variants, causal variants governing the activity of regulatory region remain to be pinpointed." to "Recent studies have employed H3K27ac to annotate many regulatory elements in pigs, narrowing down the genome regions containing candidate variants associated with complex traits identified by GWAS¹⁻³. However, causal variants hiding in these candidate variants and governing the activity of regulatory elements remain to be pinpointed." (main text: **lines 56-59**).

The introduction section should clearly state the study's purpose and rationale. Why are the new data and analyses important? What limitations existed in previous studies that this research addresses? Some of the detailed numerical results, which are duplicated in the results section, could be omitted from the introduction.

Responses: We thank Reviewer #2 for this constructive suggestion.

The study's purpose and rationale can be summarized as follows:

(1) The integration of H3K27ac quantitative trait loci (acQTLs) and expression quantitative trait loci (eQTLs) has been employed in humans to fine-map GWAS loci, yet comparable efforts in pigs remain lacking. We sought to profile active regulatory elements using H3K27ac and identify acQTLs for facilitating the dissection of the molecular mechanisms underlying pig complex traits.

(2) Genetic variants can influence complex traits by modulating gene expression with the assistance of changes to regulatory element activity, yet the extent to which genetic effects on gene expression through changes to regulatory element activity remain incompletely characterized. Here, we identified eQTLs and investigated their colocalization with acQTLs to illuminate the primary mechanism by which regulatory elements act.

(3) To demonstrate the utilization of our data in fine-mapping GWAS signals, we took advantage of liver-related traits (dulcitol, phosphatidylcholine (16:0/16:0), and published phenotypes) for prioritizing likely causal variants and identifying novel sub-threshold loci. These processes would provide a reference for improving the detection of causal variants associated with complex traits using our data.

The importance of our new data and analyses can be summarized as follows:

(1) Our H3K27ac ChIP-seq with a large sample size enables the identification of abundant new regulatory elements, expanding the pig liver regulatory element repertoire and facilitating the characterization of inter-individual variation in regulatory element activity, which helps facilitate subsequent acQTLs mapping.

(2) All data were generated using a specially designed heterogeneous pig population managed under the same external environment, which can reduce the variance of environmental factors and amplify the genetic effect.

(3) By QTLs mapping, we identified many reliable acQTLs and eQTLs for interpreting GWAS signals. By causal inference for H3K27ac and gene expression, we refer that about half of H3K27ac exhibit a concomitant rather than causative relationship with gene expression. By colocalization analysis, we identified many peak-gene pairs, facilitating the identification of likely causal genes responsible for complex traits. By analyzing the allelic imbalance of H3K27ac and transcription factors prediction, we explored the regulatory mechanism of acQTLs for H3K27ac. By deciphering GWAS signals of liver-related traits with the assistance of our data, we demonstrated their utilization in prioritizing likely causal variants and identifying novel sub-threshold loci of complex traits.

The limitations that existed in previous studies that this research addresses can be summarized as follows:

(1) While numerous studies have employed different tissue and different histone modifications to annotate regulatory elements in pigs¹⁻³, systematic identification of histone quantitative trait loci (hQTLs) that affect the activity of regulatory elements, such as acQTLs, is still lacking. In this research, we identified 28,425 acQTLs in pigs.

(2) Although acQTLs have been identified in previous human studies, the limited sample sizes and heterogeneity for samples potentially limit acQTLs identification. In this research, we used 278 pigs managed under the same external environment for acQTLs mapping, which eliminates the concerns of sample sizes and heterogeneity for samples.

(3) H3K27ac is one of the most widely studied histone modifications that serve as indicators of regulatory elements, but there is no study investigating the causal impact of H3K27ac on gene expression on a genome-wide scale. In this study, we found that about half of H3K27ac exhibit a concomitant rather than causative relationship with gene expression by causal inference analysis.

Besides, we deleted these numerical results in the Introduction section, which are duplicated in the Results section, such as lines 65, 67, 76, 77 vs. lines 90, 112, 153, 210,

211, 331, and 332 in the original manuscript. Taken together, we reworked the last two paragraphs of the introduction section (main text: **lines 49-85**).

The manuscript would benefit from incorporating findings from human and mouse studies in the results and discussion sections for comparison. For instance, in Line 158, it would be helpful to know the corresponding percentage in humans or mice. Additionally, in Line 161, it would be valuable to discuss whether the observations are conserved across species.

Responses: We thank Reviewer #2 for this very insightful comment. In Line 158 in the original manuscript, the result shows that the cis-acQTLs made up 87% of acQTLs. A similar result was obtained through ATAC-seq analysis using mouse liver conducted by Keele et al.¹¹. They found local chromatin accessibility quantitative trait loci (caQTL) made up 90% of caQTL, which is comparable to our result. Thus, we changed the sentence "The cis-acQTLs made up 87% of acQTLs." to "The majority of acQTLs are cis-acQTLs (87%), which is similar to the result of chromatin accessibility QTLs in mouse¹¹." (main text: **lines 177-178**).

In Line 161 in the original manuscript, the result shows that the cis-acQTLs have a higher enrichment in enhancer peaks than in other genomic features. A similar work conducted by Currin et al¹². shows that caQTLs are significantly enriched in enhancers (OR = 2.9) and promoters (OR = 2.0) in the human liver, and the degree of enrichment was higher for enhancers. Therefore, we changed the sentence "We found the cis-acQTLs have a higher enrichment in enhancer peaks than in other genomic features." to "We found the cis-acQTLs have a higher enrichment in enhancer peaks than in other genomic features, which is comparable to the result of chromatin accessibility QTLs in human liver¹²." (main text: **lines 180-182**).

Furthermore, we also compared our results with additional findings from human studies as follows:

(1) In Line 240 in the original manuscript, the result shows that the majority ($\pi_1=0.84$) of promoter-acQTLs were preserved in eQTLs regulating gene expression. A study conducted by Li et al.¹³ shows that the average sharing value (π_1) from promoter-acQTLs to gene expression is about 0.8 (measured by Figure S2C from their study) in Yoruba lymphoblastoid cell lines, which is consistent with our result. Thus, we changed the sentence "The majority ($\pi_1=0.84$) of promoter-acQTLs were preserved in eQTLs regulating gene expression." to "The majority ($\pi_1=0.84$) of promoter-acQTLs were preserved in eQTLs regulating gene expression, which was in line with the result in humans¹³." (main text: **lines 263-264**). Besides, we discussed the result in the Discussion section (main text: **lines 452-455**).

(2) In Line 152 in the original manuscript, the result shows that the majority of acQTLs were responsible for one peak. A previous work conducted by Grubert et al.¹⁴, showed local histone QTLs are frequently associated with one regulatory element in humans. In the revised manuscript, we changed the sentence "The result showed that the majority of acQTLs were responsible for one peak." to "The result showed that the

majority of acQTLs were responsible for one peak, which was consistent with findings in humans¹⁴." (main text: **lines 170-172**).

The discussion should add at least three more paragraphs to discuss the implications and current knowledge in different species. Then readers could understand the importance of new data and analyses.

Responses: We thank Reviewer #2 for this critical comment. We have heavily revised the Discussion section to interpret our result and discuss implications across species. Overall, we discuss three points and the details are as follows:

(1) We discussed the variability of enhancers across species, tissues, and individuals.

"The distribution of liver H3K27ac with respect to genomic features exhibits a similar pattern across pigs, humans, and mouse^{15,16}. Previous studies demonstrated that enhancers are less conserved across species and tissues compared to promoters^{17,18,3,8,19}. Our result further showed that enhancers have a lower likelihood of being shared across individuals than promoters. Comparative regulatory genomic analysis in 20 mammalian species has revealed the rapid evolution of enhancers, and the rate of divergence for enhancers is estimated to be 3 times faster than for promoters¹⁷. Newly evolved enhancers showed high inter-individual variability and tended to be less integrated in transcriptional networks⁶. Indeed, abundant enhancers without contacting promoters do not regulate gene expression¹⁸. Besides, many enhancers are functionally redundant or have modest effects on target gene expression^{20,21}. Consequently, the high variation of enhancers across species, tissues, and individuals might be better tolerated." (main text: **lines 399-410**).

(2) We discussed the potential significance of high H3K27ac signals across species, tissues, and individuals.

"Our results indicated a strong peak signal tends to have a high peak occurrence across individuals. There are two reasons for this: 1) strong peak signal is easy to detect, in turn, leading to their high occurrence; 2) regulatory elements with strong peak signals exert an essential function in pig liver, reflected by frequent occurrence. Comparably, promoters and enhancers that are active in both humans and mice have stronger H3K27ac signals than species-specific regulatory elements²². We further obtained chromatin states conducted by Pan et al⁸, and found that all-tissue shared promoters/enhancers have higher activity than liver-specific promoters/enhancers (Supplementary Figure S9). These findings suggested that regulatory elements with high H3K27ac signals tend to be stable across individuals, tissues, and species." (main text: **lines 411-420**).

(3) Combining our results with a study investigating the mechanism of initiating transcription in mice, we discussed the regulatory potential of promoters and

enhancers in modulating gene expression.

"In mouse neurons, massively parallel reporter assays demonstrated the sufficiency of promoters to independently initiate transcription while enhancers stimulate transcriptional initiation in a promoter-dependent manner²³. Analogously, our study also revealed that putative regulatory elements controlling gene expression prefer to be promoters by high enrichment of eQTLs in promoters, similar distribution between eQTLs and promoter-acQTLs, and significant colocalization between promoter-acQTLs and eQTLs." (main text: **lines 456-461**).

Overall, these revisions will enhance the manuscript's clarity and make it more informative for readers.

Finally, we would like to express our gratitude to Reviewer #2 for reviewing our manuscript and providing constructive comments, which have helped us improve the quality and clarify the meaning of our work. We hope that the responses provided here addressed all the comments in a satisfactory manner.

Responses to Reviewer #3

General comments:

Ling et al. collected large-scale omics data, including WGS, RNA-seq and ChIP-seq of H3K27ac, from 292 pig livers and then explored the genetic control of both H3K27ac activity and gene expression, as well as applied these findings to dissect genetics of complex traits in pigs. The manuscript is well-written, and the data and results are novel and valuable in the field of pig genetics and genomics as claimed by the authors. However, to make the data more impactful and allow others to reproduce the findings, I suggest that all the raw data and the matched metadata should be publicly available. I have a few technical comments and suggestions below, and hopefully they are helpful to improve the manuscript.

Responses: We sincerely thank Reviewer #3 for the thorough evaluation of our work and very positive assessment. As for data availability, we have applied for a project number (PRJCA022379) in the China National Center for Bioinformation (CNCB, <https://www.cncb.ac.cn/>) to deposit our data. All sequencing data are being uploaded.

Specific comments:

1) In figure 1a, it would be better to log-transform the peak width in the x-axis, as it is a highly right skewed distribution. Then you might see that the frequency of peaks is also decreases as their width decreases (Line 89-90).

Responses: We thank Reviewer #3 for this good suggestion. We performed a log₂ transformation on the x-axis of Figure 1A (**Fig.R1**). It is still a right-skewed distribution but to a lesser extent. Besides, we did observe the frequency of peaks decrease as their width decreases in the left after log₂ transformation. Thus, we corrected Figure 1A in the revised manuscript and changed the sentence "The average peak width across all samples was 691 bp, and the frequency of peaks decreases as their width increases" to "The average peak width across all samples was 691 bp, and the frequency distribution of peaks is highly right-skewed." (main text: **lines 93-94**).

Fig.R1 Distribution of H3K27ac peak width

2) Line250, what is the cutoff of Spearman correlation used to link peaks to genes?

Responses: We thank Reviewer #3 for this critical comment. To reduce the false discovery rate in the identification of peak-gene pairs by colocalization analysis, we calculated the Spearman's correlation between peak abundance and gene expression to filter unrelated peak-gene pairs. As multiple statistic tests were conducted, we employed the Benjamini-Hochberg method to perform multiple testing corrections and adjust the nominal P -value²⁴. The adjusted P -value, also called Benjamini-Hochberg's q -value, was used to determine the significance of Spearman's correlation with a cutoff 0.05. The "p.adjust(P -value, method=BH)" function implemented in R software was used to perform the adjustment. In the revised manuscript, we have corrected the sentence "To reduce the false discovery rate, we kept peak-gene pairs in which the peak was the promoter peak of this gene or the peak activity correlated with gene expression (Spearman's correlation, Benjamini-Hochberg's q -value < 0.05)" to "To reduce the false discovery rate, we kept peak-gene pairs in which the peak is the promoter peak of this gene, or the peak activity is significantly associated with gene expression by the Spearman's correlation method after multiple testing correction of Benjamini-Hochberg (adjust P -value < 0.05)." (main text: **lines 272-275**).

3) Before QTL mapping, it would be good to look at the cis/trans-heritability of these H3K27ac peaks and any differences between promoter-H3K27ac and enhancer-H3K27ac.

Responses: We thank Reviewer #3 for this constructive comment. Insights into the inter-individual variation of peak activity and its heritability (h^2) can help in understanding pathways from DNA to H3K27ac. For each peak, we estimated cis-heritability (h^2_{cis}) using variants within ± 1 Mb from the peak, while trans-heritability (h^2_{trans}) was computed using variants beyond ± 5 Mb from the peak. We estimated the heritability of 88,926 peak activity in autosomes using 278 individuals. Among these peaks, approximately 10%/5% peaks have h^2_{cis}/h^2_{trans} greater than 0.2 (**Supplementary Figure S2A and S2B**). Mean h^2_{cis} was 0.064, which was significantly higher than h^2_{trans} (mean=0.029, T-test, P -value < 2.2×10^{-16}). We further group h^2 into promoters- h^2 and enhancers- h^2 . Estimates of promoters- h^2 showed significantly higher than enhancers- h^2 , regardless of the cis or trans patterns (**Supplementary Figure S2C and S2D**). Besides, estimates of h^2_{cis} are still higher than h^2_{trans} in both groups (**Supplementary Figure S2E and S2F**).

We have added the above content to the Result section of the revised manuscript as follows: "Insights into the inter-individual variation of peak activity and its heritability (h^2) can help in understanding pathways from DNA to H3K27ac. We estimated the heritability of peak activity of 88,926 peaks located in autosomes using 278 individuals. The genomes of these individuals were sequenced to an average depth of $7.8 \times 25,26$. The activity of regulatory elements is controlled by both cis-QTL and trans-QTL¹¹. Thus, we estimated for each peak h^2_{cis} (the variance explained by genetic variants

located within ± 1 Mb from the peak) and h^2_{trans} (the variance explained by genetic variants located beyond ± 5 Mb from the peak). Among 88,926 peaks, 10% peaks have h^2_{cis} greater than 0.2, and 5% peaks have h^2_{trans} greater than 0.2 (**Supplementary Figure S2A and S2B**). Mean h^2_{cis} was 0.064, which was significantly higher than h^2_{trans} (mean=0.029, T-test, P -value $< 2.2 \times 10^{-16}$). We further group h^2 into promoters- h^2 and enhancers- h^2 . Estimates of promoters- h^2 showed significantly higher than enhancers- h^2 , regardless of the cis or trans patterns (**Supplementary Figure S2C and S2D**). Besides, estimates of h^2_{cis} are still higher than h^2_{trans} in both groups (**Supplementary Figure S2E and S2F**). (main text: **lines 145-158**).

We have added the analysis process to the Materials and Methods section of the revised manuscript as follows: "We employed the GREML-LDMS-I method^{27,28} for heritability estimation. For each peak, we estimated cis-heritability using variants within ± 1 Mb, while trans-heritability was computed using variants located beyond ± 5 Mb. To summarize, we first calculated the segment-based LD score with a 200 Kb window size. SNPs were then stratified into four groups according to LD score quartiles. Following this, GRMs were generated for each SNP group and utilized to estimate heritability." (main text: **lines 625-630**).

Moreover, we discussed the result in the Discussion section of the revised manuscript as follows: "The estimates of heritability showed genetic variants located within cis windows are more likely to affect peak activities than trans variants, supporting the result that most acQTLs were cis-acQTLs." (main text: **lines 434-436**)

Supplementary Figure S2. Estimates of heritability of H3K27ac peaks. **(A)** Distribution of cis-heritability for all peaks (left), promoter peaks (middle), and enhancer peaks (right). The average estimates of cis-heritability were shown with the red dashed line. **(B)** Distribution of trans-heritability for all peaks (left), promoter peaks (middle), and enhancer peaks (right). The average estimates of trans-heritability were shown with the red dashed line. **(C)** Comparison of cis-heritability for enhancer and promoter peaks. **(D)** Comparison of trans-heritability for enhancer and promoter peaks. **(E)** Comparison of cis-heritability and trans-heritability for promoter peaks. **(F)** Comparison of cis-heritability and trans-heritability for enhancer peaks. The boxplots display the median, the 25th and 75th percentiles. The whiskers indicate the minimum and maximum values, and outliers are shown as points outside the ends of the whiskers.

4) For trans-QTL mapping, which covariates have been accounted for, same as cis-QTL mapping? Trans-QTL often have small effects which can be contaminated by artifacts easily (e.g., subtle population structure and cell type composition) like normal GWAS of complex traits. So, it could be better to use much more stringent cutoffs and methods e.g., a linear mixed model, to see if the results are still stable.

Responses: We thank Reviewer #3 for this very important comment. For cis-QTLs mapping, we employed the FastQTL software used in GTEx eQTL pipeline (<https://github.com/broadinstitute/gtex-pipeline/tree/master/qtl>) to discover QTLs in cis with the standard permutation pass workflow (design for cis-QTLs mapping, <https://qtltools.github.io/qtltools/>). In the workflow, we corrected for sex, slaughter age, slaughter batch, RNA Integrity Number (RIN) values (not for acQTLs mapping), sequencing read counts, the first three principal components of genotypes, and 45 PEER factors. For trans-QTLs mapping, the covariates we included were the same as those for cis-QTLs mapping. The approximate pass workflow (design for trans-QTLs mapping, <https://qtltools.github.io/qtltools/>) implemented in QTLtools software was employed to discover QTLs in trans.

The significance of cis-QTLs and trans-QTLs mapping was determined through at least 1000 permutations. The false discovery rate (FDR) threshold was set to 0.05 in cis-QTLs and trans-QTLs mapping. Because the number of statistic tests for trans-QTLs mapping is greater than that of cis-QTLs mapping, the cutoffs for trans-QTLs mapping are much more stringent than that for cis-QTLs mapping if the nominal *P*-value is used. Besides, we have corrected for principal components of genotypes and PEER factors in the trans-QTLs mapping, which would partly eliminate contaminations of artifacts such as subtle population structure and cell type composition.

To determine the reliability of the trans-QTL mapping results, we employed a mixed linear model to identify trans-QTLs with the fastGWA tool²⁹ in GCTA software. The results showed that 4 of 2,208 trans-eQTLs and 1 of 3,589 trans-acQTLs did not exceed the empirical significance threshold of 5×10^{-8} , indicating the robustness of trans-QTL identification. These results have been added to the Materials and Methods section as follows: "To determine the reliability of the trans-eQTL mapping results, we employed a mixed linear model to identify trans-eQTLs with the fastGWA tool²⁹ in GCTA software. The results showed that 99% of trans-eQTLs reach the empirical significance threshold of 5×10^{-8} , indicating the robustness of trans-eQTLs identification." (main text: **lines 653-657**). The corresponding *P*-values from the mixed linear model have been added to Table S5 and Table S11.

5) For H3K27ac QTL mapping, how did the authors get 45 PEER factors, same as eQTL mapping, using gene expression rather than H3K27ac signals?

Responses: We thank Reviewer #3 for this critical comment. We apologize for not clarifying the analysis process of acQTL mapping. PEER (probabilistic estimation of expression residuals) factors represent unmeasured and unknown confounders in eQTL

mapping, which can be predicted by PEER software³⁰. We employed this method for eQTL mapping, as well as for acQTL mapping. The PEER factor estimates were also applied to ChIP-seq data to prevent the potential effects of unknown confounders^{15,31}. In our study, the PEER factors we used for acQTL mapping were produced based on H3K27ac signals. Following the recommendations from the GTEx Consortium guidelines, when the sample size ranges between 250 to 350, the number of PEER factors of eQTLs can be set to 45 (as detailed at <https://github.com/broadinstitute/gtex-pipeline/tree/master/qtl>). Given that the H3K27ac samples originate from the same population and are likely subject to consistent confounding factors, we proposed that the number of PEER factors for H3K27ac should align with that used for gene expression analysis. We rework the acQTL mapping process in the Materials and Methods section of revised manuscript as follows: 1) " PEER (probabilistic estimation of expression residuals) factors represent unmeasured and unknown confounders in eQTL mapping, which can be predicted by PEER software³⁰" (main text: **lines 637-639**); 2) "Similar to eQTL mapping, PEER factors were predicted based on H3K27ac signals and the number of factors was also set to 45." (main text: **lines 663-665**).

6) For allele imbalance analysis of acQTLs, how did the authors take account of mapping bias?

Responses: We thank Reviewer #3 for this insightful comment. To mitigate the effects of mapping bias of reads, we employed the WASP software developed by van de Geijn et al. to remove reads exhibiting allele-biased mapping³². Briefly, WASP was used to flag reads that should be remapped due to overlapping with genetic variants. The alleles of variants were then computationally swapped, and the reads were remapped to determine if they would still be aligned to the original location. After alleles swap, reads with unchanged mapping genome positions were retained. Consequently, twenty-one of 32 previous lead acQTL variants meet our inclusion criteria and can be used for examining the allelic imbalance of the H3K27ac activity.

Thus, we have changed the sentence "To validate the causality of these lead acQTL variants, we examined the allelic imbalance of the H3K27ac activity by 32 lead acQTL variants, with sufficient heterozygous samples, that is inside target peaks and with PPs exceeding 0.9. We observed that 23 lead acQTL variants exhibited consistency between acQTL analysis and allelic imbalance analysis in terms of effect allele direction, 15 of which showed significant differences between reads coverage of reference alleles and alternative alleles. To confirm these 15 lead acQTL variants further, we retrieved 24 H3K27ac data of pig livers from three independent studies. Seven out of 15 lead acQTL variants had sufficient heterozygous individuals for the statistical test. Six lead acQTL variants showed a consistent tendency, and three were successfully verified" to "To validate the causality of these lead acQTL variants, we examined allelic imbalance of H3K27ac activity for 21 lead acQTL variants that met the following criteria: 1) reads covering the variants have no mapping bias³²; 2) a sufficient number of heterozygous samples for the test; 3) the variants are located inside H3K27ac peaks; 4) the PPs of

variants exceeding 0.9. We observed that 14 lead acQTL variants exhibited consistency between acQTL analysis and allelic imbalance analysis in terms of effect allele direction, eight of which showed significant differences between reads coverage of reference alleles and alternative alleles (Supplementary Table S6). To confirm these 8 lead acQTL variants further, we retrieved 24 H3K27ac data of pig livers from three independent studies¹⁻³. Five out of 8 lead acQTL variants had sufficient heterozygous individuals for the statistical test. Four lead acQTL variants showed a consistent tendency, and 2 were successfully verified." (main text: **lines 190-200**).

In the revised manuscript, we have added the corresponding sentences to the Materials and Methods section of the revised manuscript as follows: "To mitigate the effects of mapping bias of reads, we employed the WASP software developed by van de Geijn et al. to remove reads exhibiting allele-biased mapping³². Briefly, WASP was used to flag reads that should be remapped due to overlapping with genetic variants. The alleles of variants were then computationally swapped, and the reads were remapped to determine if they would still be aligned to the original location. After alleles swap, reads with unchanged mapping genome positions were retained." (main text: **lines 681-687**). Supplementary Table S6, Figure 2, and Supplementary Figure S2 were updated correspondingly.

7) For variance decomposition of gene expression, did the authors only consider the cis variants and H3K27ac signals, and how the genetic relatedness of individuals was controlled?

Responses: We thank Reviewer #3 for this important comment and apologize for not clarifying this in the Materials and Methods section of the revised manuscript. We used a linear mixed model implemented in LIMIX software^{33,34} to investigate the contributions to gene expression variability from cis variants and H3K27ac signals. The model is as follows:

$$y = N(1\mu, \sigma_l^2 K_l + \sigma_g^2 K_g + \sigma_h^2 K_h + \sigma_e^2 I)$$

Where y represents the gene expression levels across all samples, 1μ represents an offset term, K_l is relatedness matrix built by cis genetic variants or H3K27ac signals, K_g represents a relationship matrix considering all variants and $\sigma_e^2 I$ is the noise term.

K_h represents expression heterogeneity and is calculated using the equation $K_h = (1/G)ZZ^T$, in which Z is the $N \times G$ gene expression matrix for N samples and G genes. Thus, for variance decomposition, we considered cis variants, H3K27ac signals, expression heterogeneity, and relationship matrix. The genetic relatedness of individuals was controlled by the relationship matrix (K_g) in the linear mixed model.

In the revised manuscript, we have added the details of the variance decomposition of gene expression to the Materials and Methods section of the revised manuscript (main text: **lines 725-741**).

8) The methods for the causal inference of H3K27ac and gene expression are too simple, e.g., what cutoffs have been used to define significance level.

Responses: We thank Reviewer #3 for pointing out this. Details of the causal inference of H3K27ac and gene expression were added to the Materials and Methods section of the revised manuscript as follows:

"The causal inference test (CIT) is a mediation-based method introduced by Millstein et al.³⁵, which examines the hypothesis that a potential causal mediator (G, such as H3K27ac signal) mediates a causal association between a genetic locus (L) and a quantitative trait (T, such as gene expression). Causality (from genetic variants to the mediator to the trait) can be inferred if four conditions are met:

- (1) L and G are associated
- (2) L and T are associated
- (3) L is associated with G, given T
- (4) L is independent of T, given G

A total of 1,900 candidate L/G/T trios meeting the first two conditions, obtained from peak-gene colocalization analysis, were used for CIT, which can test the strength of a chain of mathematical conditions that as a set are consistent with causal mediation. The Intersection-Union Test framework³⁵ is used to compute an omnibus *P*-value for the suite of conditions that would function as CIT. For each particular trio with genotype and gene/H3K27ac levels, CIT outputs omnibus *P*-values of a causal model (genetic variants → H3K27ac signals → gene expression; pCausalCIT) and a reactive model (genetic variants → gene expression → H3K27ac signals; pReactiveCIT), which represent the highest *P*-value (i.e., minimal significance) among the four component tests. The CIT predicted casual direction when pCausalCIT < 0.05 and pReactiveCIT > 0.05 (Type1), and reactive direction when pCausalCIT > 0.05 and pReactiveCIT < 0.05 (Type2). Trios with pCausalCIT > 0.05 and pReactiveCIT > 0.05 were considered independent (Type3). The CIT makes no call if pCausalCIT < 0.05 and pReactiveCIT < 0.05 (Type0)" (main text: **lines 744-765**).

9) For colocalization analysis, there are some often-used methods e.g., COLOC. Did the authors try some of them instead of just looking at LD?

Responses: We thank Reviewer #3 for this critical comment. To strengthen the reliability of our results, we combined LD analysis with the Bayesian test implemented in COLOC software to perform colocalization analysis⁴. For 2,299 peak-gene pairs identified by LD analysis, we colocalized their QTLs by COLOC software with a posterior probability greater than 0.8³⁶. The result showed that 1,818 peak-gene pairs were connected by COLOC and LD analysis. Consequently, we updated the related results in the revised manuscript as follows:

(1) In the Results section, we have changed the sentence "In total, we identified 1,394

target genes for 1,928 peaks, comprising 2,299 unique peak-gene connections. Most (82%) of the peaks were linked to one gene." to "Besides, we employed the Bayesian test to perform colocalization and intersected the above results⁴. In total, we identified 1,183 target genes for 1,616 peaks, comprising 1,818 unique peak-gene connections. Most (90%) of the peaks were linked to one gene." (main text: **lines 275-278**).

(2) In the results of metabolism-related GWAS analysis, we have added the corresponding PP4 values of colocalization for GWAS QTL vs. eQTL, GWAS QTL vs. acQTL, and acQTL vs. eQTL (main text: **lines 317-321**).

(3) In the results of interpreting public GWAS data of pig trait, we have changed the sentence "We linked 64 phenotypes to 258 gene-peak pairs via LD score ($r^2 > 0.8$), resulting in 297 candidate variants." to "We linked 55 phenotypes to 111 gene-peak pairs via LD score ($r^2 > 0.8$), resulting in 167 candidate variants. ". (main text: **lines 365-367**).

(4) We have added the corresponding content to the Materials and Methods section: "We employed the Bayesian test implemented in COLOC software to assess colocalization⁴. All variants within a ± 1 Mb from the lead variants of eQTLs and acQTLs were intersected and used for colocalization. The threshold of posterior probabilities of H4 (Association with H3H27ac peaks and gene expression, one shared genetic variant) (PP4) was set to 0.8." (main text: **lines 780-783**).

(5) Table S13, Table S14, Table S15, Figure 3D, and Figure 3F were updated accordingly.

Finally, we sincerely thank Reviewer #3 for providing insightful comments after reviewing our manuscript, which have helped us significantly improve the quality and clarity of our work. We hope that our responses have addressed all of the concerns in a satisfactory manner.

Reference

- 1 Zhu, Y. *et al.* Mapping and analysis of a spatiotemporal H3K27ac and gene expression spectrum in pigs. *Sci China Life Sci* **65**, 1517-1534,(2022).
- 2 Zhao, Y. *et al.* A compendium and comparative epigenomics analysis of cis-regulatory elements in the pig genome. *Nat. Commun.* **12**, 2217,(2021).
- 3 Kern, C. *et al.* Functional annotations of three domestic animal genomes provide vital resources for comparative and agricultural research. *Nat. Commun.* **12**, 1821,(2021).
- 4 Giambartolomei, C. *et al.* Bayesian test for colocalisation between pairs of genetic association studies using summary statistics. *PLoS Genet.* **10**, e1004383,(2014).
- 5 Chandra, V. *et al.* Promoter-interacting expression quantitative trait loci are enriched for functional genetic variants. *Nat. Genet.* **53**, 110-119,(2021).
- 6 Castelijns, B. *et al.* Recently evolved enhancers emerge with high interindividual variability and less frequently associate with disease. *Cell Rep.* **31**, 107799,(2020).
- 7 Light, N. *et al.* Interrogation of allelic chromatin states in human cells by high-density ChIP-genotyping. *Epigenetics* **9**, 1238-1251,(2014).
- 8 Pan, Z. *et al.* Pig genome functional annotation enhances the biological interpretation of complex traits and human disease. *Nat. Commun.* **12**, 5848,(2021).
- 9 Foissac, S. *et al.* Multi-species annotation of transcriptome and chromatin structure in domesticated animals. *BMC Biol.* **17**, 108,(2019).
- 10 Bailey, T. L. *et al.* MEME SUITE: tools for motif discovery and searching. *Nucleic Acids Res.* **37**, W202-W208,(2009).
- 11 Keele, G. R. *et al.* Integrative QTL analysis of gene expression and chromatin accessibility identifies multi-tissue patterns of genetic regulation. *PLoS Genet.* **16**, e1008537,(2020).
- 12 Currin, K. W. *et al.* Genetic effects on liver chromatin accessibility identify disease regulatory variants. *Am. J. Hum. Genet.* **108**, 1169-1189,(2021).
- 13 Li, Y. I. RNA splicing is a primary link between genetic variation and disease. *science*,(2016).
- 14 Grubert, F. *et al.* Genetic Control of Chromatin States in Humans Involves Local and Distal Chromosomal Interactions. *Cell* **162**, 1051-1065,(2015).
- 15 Caliskan, M. *et al.* Genetic and Epigenetic Fine Mapping of Complex Trait Associated Loci in the Human Liver. *Am. J. Hum. Genet.* **105**, 89-107,(2019).
- 16 Lopez-Perez, A., Remeseiro, S. & Hornblad, A. Diet-induced rewiring of the Wnt gene regulatory network connects aberrant splicing to fatty liver and liver cancer in DIAMOND mice. *Sci. Rep.* **13**, 18666,(2023).
- 17 Villar, D. *et al.* Enhancer evolution across 20 mammalian species. *Cell* **160**, 554-566,(2015).
- 18 Vangala, P. *et al.* High-resolution mapping of multiway enhancer-promoter interactions regulating pathogen detection. *Mol. Cell* **80**, 359-373. e358,(2020).
- 19 Hariprakash, J. M. & Ferrari, F. Computational biology solutions to identify enhancers-target gene pairs. *Comput. Struct. Biotechnol. J.* **17**, 821-831,(2019).
- 20 Choi, J. *et al.* Evidence for additive and synergistic action of mammalian enhancers during cell fate determination. *eLife* **10**, e65381,(2021).
- 21 Osterwalder, M. *et al.* Enhancer redundancy provides phenotypic robustness in mammalian development. *Nature* **554**, 239-243,(2018).
- 22 Donnard, E. *et al.* Comparative analysis of immune cells reveals a conserved regulatory lexicon.

- Cell Syst.* **6**, 381-394. e387,(2018).
- 23 Nguyen, T. A. *et al.* High-throughput functional comparison of promoter and enhancer activities. *Genome Res.* **26**, 1023-1033,(2016).
- 24 Benjamini, Y. & Hochberg, Y. Controlling the false discovery rate: a practical and powerful approach to multiple testing. *J R Stat Soc Series B Stat Methodol* **57**, 289-300,(1995).
- 25 Yang, H. *et al.* ABO genotype alters the gut microbiota by regulating GalNAc levels in pigs. *Nature* **606**, 358-367,(2022).
- 26 Zhang, Y. *et al.* Subcutaneous and intramuscular fat transcriptomes show large differences in network organization and associations with adipose traits in pigs. *Sci China Life Sci* **64**, 1732–1746,(2021).
- 27 Evans, L. M. *et al.* Comparison of methods that use whole genome data to estimate the heritability and genetic architecture of complex traits. *Nat. Genet.* **50**, 737-745,(2018).
- 28 Yang, J. *et al.* Genetic variance estimation with imputed variants finds negligible missing heritability for human height and body mass index. *Nat. Genet.* **47**, 1114-1120,(2015).
- 29 Jiang, L. *et al.* A resource-efficient tool for mixed model association analysis of large-scale data. *Nat. Genet.* **51**, 1749-1755,(2019).
- 30 Stegle, O., Parts, L., Piipari, M., Winn, J. & Durbin, R. Using probabilistic estimation of expression residuals (PEER) to obtain increased power and interpretability of gene expression analyses. *Nat Protoc* **7**, 500-507,(2012).
- 31 Waszak, S. M. *et al.* Population Variation and Genetic Control of Modular Chromatin Architecture in Humans. *Cell* **162**, 1039-1050,(2015).
- 32 Van De Geijn, B., McVicker, G., Gilad, Y. & Pritchard, J. K. WASP: allele-specific software for robust molecular quantitative trait locus discovery. *Nat. Methods* **12**, 1061-1063,(2015).
- 33 Casale, F. P., Rakitsch, B., Lippert, C. & Stegle, O. Efficient set tests for the genetic analysis of correlated traits. *Nat. Methods* **12**, 755-758,(2015).
- 34 Chen, L. *et al.* Genetic Drivers of Epigenetic and Transcriptional Variation in Human Immune Cells. *Cell* **167**, 1398-1414 e1324,(2016).
- 35 Millstein, J., Zhang, B., Zhu, J. & Schadt, E. E. Disentangling molecular relationships with a causal inference test. *BMC Genet.* **10**, 23,(2009).
- 36 Roychowdhury, T. *et al.* Genome-wide association meta-analysis identifies risk loci for abdominal aortic aneurysm and highlights PCSK9 as a therapeutic target. *Nat. Genet.* **55**, 1831-1842,(2023).

REVIEWERS' COMMENTS:

Reviewer #1 (Remarks to the Author):

In my opinion, all my comments and those of the other reviewers have been very thoroughly addressed by the authors. I do not have any further comments.

Reviewer #2 (Remarks to the Author):

After a thorough review of the revised manuscript COMMSBIO-23-3369A titled "Omics-based construction of regulatory variants and its application in deciphering pig liver-related traits," I am pleased to acknowledge the comprehensive revisions made by the authors in response to the initial comments and suggestions. The authors have adeptly addressed all the previously raised concerns, making substantial improvements to the manuscript. The modifications and additions significantly enhance the clarity, depth, and scientific rigor of the study.

Reviewer #3 (Remarks to the Author):

Thanks for the responses, and all my concerns have been addressed.